# ✿CURV: Enhancing Chart Understanding through Grounded Visual Reasoning

## Abstract

Chart question answering (CQA) requires multimodal large language models (MLLMs) to integrate visual comprehension with logical reasoning, yet current models struggle with accurate visual grounding and coherent reasoning chains. While extrinsic chain-of-thought prompting and visual cues significantly improve performance, current MLLMs lack intrinsic visually grounded reasoning capabilities, leading to inaccurate perception and reasoning disconnected from visual evidence. To address these limitations, we propose ✿CURV, a curriculum learning framework that develops intrinsic grounded visual reasoning capabilities by reformulating CQA as multi-turn visual reasoning, where each step coordinates logical reasoning with dynamic visual grounding through spatial attention concentration. To assist model learning, we further introduce CCQA, a three-level curriculum dataset with scalable synthetic generation across diverse chart types and reasoning patterns. Our curriculum systematically progresses from basic single-operation reasoning to complex multi-chart compositional tasks. Experiments demonstrate that ✿CURV achieves up to $10.79\%$ accuracy improvements over baselines and strong generalization to real-world benchmarks and out-of-domain multimodal reasoning tasks, validating the effectiveness of internalizing visual reasoning with dynamic grounding for enhanced chart understanding capabilities.

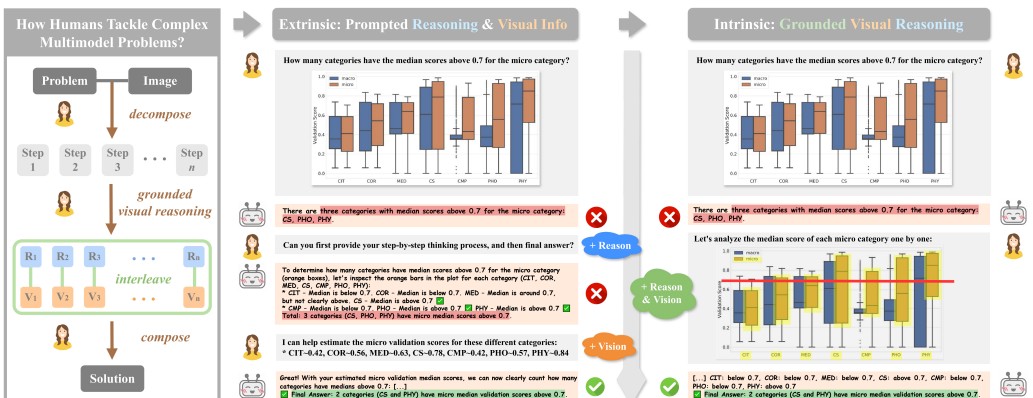

Figure 1: **From Extrinsic Assistance to Intrinsic Grounded Visual Reasoning.** Inspired by human ways of thinking, we internalize extrinsic **CoT prompting** and **visual guidance** to intrinsic capabilities, enabling models to perform **visual grounded reasoning** through dynamically shifting focuses across targeted image regions.

## 1 Introduction

*How do humans tackle multimodal problems?* Supported by cognitive theories (Baddeley et al., 1974; Johnson-Laird, 1983; Barsalou, 2008; Grant & Spivey, 2003), humans **decompose** complex tasks into stepwise reasoning chains, **interleave** each step with dynamic visual grounding, and **compose** these grounded steps into a coherent solution (Fig. 1). Chain-of-thought (CoT) reasoning has demonstrated its effectiveness in decomposing problems into step-wise inferences (Xu et al., 2024; Zhang et al., 2025a). This ability becomes more critical in multimodal reasoning, where multimodal

large language models (MLLMs) are expected to integrate visual and textual information (Fan et al., 2025; Zhang et al., 2025b) while visual perception errors contribute to the majority of multimodal reasoning failures (Wang et al., 2025c). Without external support such as explicit CoT prompting or visual cues (Fig. 1), MLLMs struggle with grounded visual reasoning (Wang et al., 2025b;a).

This challenge is particularly evident in chart question answering (CQA), where models need to faithfully interpret complex geometric structures, spatial relationships, and quantitative patterns to derive correct answers. As a result, CQA requires models to accurately perceive visual details, perform step-by-step reasoning over interconnected components, and dynamically shift focus across different chart regions (Chen et al., 2025). However, existing MLLMs exhibit several fundamental limitations in this setting (§2 & A): **(1) Decomposition:** *They struggle to decompose complex problems into coherent chains of reasoning, often producing inconsistent or logically flawed intermediate steps (Fig. 11-12)*; **(2) Interleaved Visual Reasoning:** *They show limitations in accurately grounding individual reasoning steps in the visual input, such as misreading chart values or failing to attend to the correct regions (Fig. 11)*; and **(3) Composition:** *They exhibit difficulties in integrating visual grounding and logical reasoning across multiple steps into a coherent, interleaved chain, leading to a disconnect between what is perceived, reasoned, and concluded (Fig. 12)*. Collectively, these limitations lead to inaccurate perception and reasoning that is disconnected from the visual evidence, ultimately causing errors even when the necessary information is present.

To address these limitations, we propose 🐾 **CURV**, a curriculum learning framework that develops intrinsic visually grounded reasoning capabilities in MLLMs. Our approach reformulates CQA as multi-turn reasoning processes where each step explicitly coordinates logical reasoning with dynamic visual grounding. Instead of relying on extrinsic assistance, 🐾 CURV enables models to internalize the ability through dynamically focusing on relevant chart regions while maintaining coherent reasoning chains across steps. This curriculum learning progresses from single-operation reasoning to complex multi-operation compositions, allowing models to gradually develop both visual grounding accuracy and reasoning sophistication together. Our main contributions are:

- We propose 🐾 **CURV**, a curriculum learning framework that develops intrinsic visual grounded reasoning capabilities by guiding models to progressively coordinate visual attention with logical reasoning, transitioning from basic single-operation tasks to complex nested reasoning (§3).
- We introduce a scalable synthetic CQA data generation method (§4) that systematically increases task complexity via nested reasoning chains, enabling the efficient creation of curriculum data.
- We present the *Curriculum Chart Question Answering* (**CCQA**) (§4) that supports curriculum learning across four different generation modes for comprehensive evaluation (§2 & 5).
- Our experiments demonstrate that CoT reasoning with visual grounding provides models with step-by-step alignment between visual perception and logical reasoning, leading to notable performance improvements across different chart types, task complexity, and domains (§6).

## 2 WHAT PROHIBITS MLLMs FROM CHART UNDERSTANDING SUCCESS?

### 2.1 PRELIMINARY ANALYSIS OF CQA FAILURES

*What are the bottlenecks that hinder MLLMs from correctly answering queries about chart images?*

We evaluate GPT-4o (OpenAI, 2024) on 60 CQA samples randomly selected from CharXiv (Wang et al., 2024) with four modes: **(1) Answer only (A):** Model directly generates the answer; **(2) Vision + Answer (VA):** Model is provided with human-annotated visual information to give the answer; **(3) Reasoning + Answer (RA):** Model

Table 1: **CQA Preliminary.** Preliminary results on GPT-4o using accuracy (%) and relative improvements $\Delta_{acc}$ (%). Error analysis counts the number of cases.

| Mode | Error Analysis (*count*) | | | Acc (%) | $\Delta_{acc}$ (%) |
|---|---|---|---|---|---|
| | Vision | Reasoning | Answer | | |
| A | – | – | 34 | 43.33 | – |
| VA | 0 | 18 | 5 | 61.67 | ↑ 18.33 |
| RA | 17 | 6 | 5 | 53.33 | ↑ 10.00 |
| RVA | 0 | 5 | 3 | 86.67 | ↑ **43.33** |

is prompted to first generate CoT reasoning, followed by the final answer; **(4) Reasoning + Vision + Answer (RVA):** Model is provided with the same visual information as in **VA**, prompted to first generate CoT reasoning and then the final answer. Surprisingly (Table 1), GPT-4o achieves significantly higher scores (↑ 10.00%) when prompted to generate reasoning without visual cues (**RA**). Furthermore, combining perception with reasoning (**RVA**) leads to the highest performance (↑ 43.33%). These reveal MLLMs' lack of logical decomposition and visual reasoning capabilities.

## 2.2 ENHANCING CHART REASONING THROUGH DYNAMIC VISUAL GROUNDING

*How to improve MLLMs' intrinsic visual reasoning capabilities?*

Motivated by the effectiveness of extrinsic CoT prompting and visual guidance (Tab. 1), we aim to internalize these capabilities within MLLMs (Fig. 1). To concretize this approach, we expand our preliminary exploration to visual reasoning with grounded focuses (Table 2) using CCQA (§4) and two models: Qwen2.5-VL-7B (Qwen, 2025) and GPT-4o (OpenAI, 2024). Similarly, we examine model performance under four modes using ground-truth reasoning chains and visual grounding. Despite increased curriculum difficulty ($1 \leq D \leq 3$), both models achieve higher accuracy when equipped with either reasoning or visual assistance, with

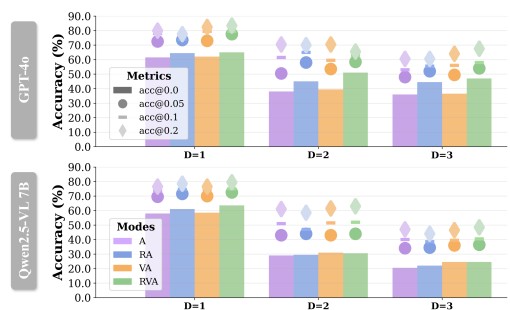

Figure 2: **Performance Across Reasoning Depths and Modes.** We evaluate CQA accuracy (%) of Qwen2.5-VL-7B and GPT-4o on 1,800 randomly selected samples from CCQA (evenly distributed across $D_i, 1 \leq i \leq 3$).

performance further heightened when both are combined. Consistently, extrinsic assistance enhances chart understanding through structured reasoning and visual grounding, motivating our core hypothesis: *MLLMs can internalize these capabilities through grounded visual reasoning (§3 & 4).*

## 3 🌸CURV: CHART REASONING WITH DYNAMIC VISUAL GROUNDING

### 3.1 PROBLEM FORMULATION

Based on our preliminary analysis (§2 & A), we identify three fundamental limitations of MLLMs (§1 & Fig. 3).

**Problem Definition.** Given a chart image $\mathcal{I} \in \mathbb{R}^{H \times W \times C}$ and a question $Q$, the goal of CQA is to generate the answer $A$. However, current MLLMs directly learn the mapping:

$$f_\theta : (\mathcal{I}, Q) \to A \tag{1}$$

Not only does this direct mapping approach lack an intermediate reasoning structure that enables accurate visual perception and robust visual understanding, but it also fails to effectively and dynamically ground reasoning chains in visual space.

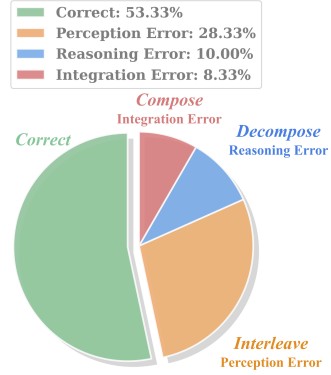

Figure 3: **Error Distribution.** Visualization of error analysis (§ 2).

**Our Approach.** We propose to decompose this problem into a multi-turn reasoning process with dynamic visual grounding. Specifically, we reformulate the CQA task as:

$$f_\theta : (\mathcal{I}, Q) \to \{(R_1, B_1), (R_2, B_2), \ldots, (R_T, B_T)\} \to A \tag{2}$$

where $R_t$ represents the $t$-th reasoning step in natural language, $B_t = (x_t, y_t, w_t, h_t)$ denotes bounding boxes that grounds $R_t$ in the visual space, $T$ is the total number of reasoning steps to reach $A$, and the sequence $\{(R_t, B_t)\}_{t=1}^T$ forms a structured progressive visual reasoning chain.

### 3.2 MULTI-TURN REASONING WITH DYNAMIC VISUAL GROUNDING

We design a multi-turn visual reasoning framework (🌸CURV) that enables models to develop intrinsic progressive reasoning capabilities with dynamic visual grounding, moving beyond extrinsic assistance toward self-contained visual reasoning for enhanced CQA performance (Fig. 1).

**Progressive Reasoning Generation.** Each reasoning step $R_t$ is conditioned on the chart image $\mathcal{I}$, the question $Q$, and the previous reasoning context:

$$(R_t, B_t) = f_\theta(\mathcal{I}, Q, \{(R_{t'}, B_{t'})\}_{t'=1}^{t-1}) \tag{3}$$

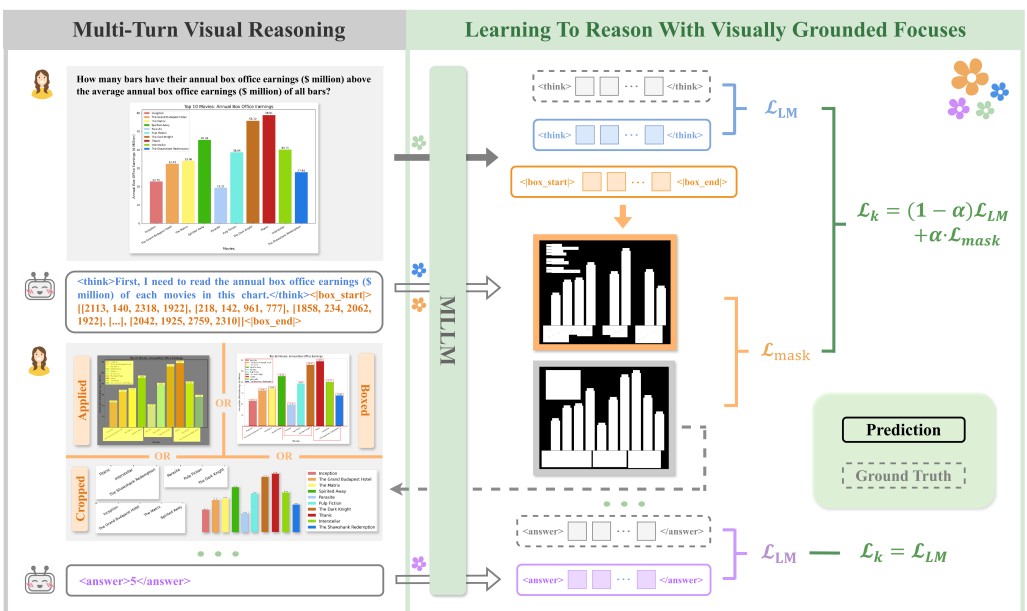

Figure 4: **Method Overview.** Proposing three visual grounding approaches (§ C), we implement **multi-turn visual reasoning** to enhance **perception** and **reasoning** capabilities.

This approach develops the model's intrinsic progressive visual reasoning capabilities, enabling it to: (1) *dynamically focus* on relevant chart regions through bounding box prediction $B_t$, (2) *generate structured reasoning* that explicitly connects visual evidence to logical steps, and (3) *maintain reasoning coherence* across multiple turns through contextual conditioning (§1).

**Visual Grounding.** We propose three visual grounding strategies, including **applied**, **boxed**, and **cropped**, to dynamically support reasoning with shifting visual focuses (Fig. 4), thereby enhancing the model's visual reasoning capacity. Our implementation details are elaborated in §C.

**Reasoning Depth.** To formalize the reasoning complexity in chart question answering, we introduce two distinct but complementary concepts that characterize the reasoning process:

- **Number of Reasoning Steps** ($T$): The total number of CoT reasoning steps $\{R_t\}_{t=1}^T$ a model goes through to reach the final answer $A$. In multi-turn reasoning, each turn represents one step $R_t$ ($t \in [1, T]$). Multiple reasoning steps may operate at the same logical complexity level (*i.e.*, *reasoning depth tier* below) while contributing different pieces of information toward the solution.
- **Tier of Reasoning Depth** ($D$): The maximum number of nested logical functions required to solve the task, corresponding to the deepest level of functional composition in the reasoning chain. Formally, for a question requiring nested functions $f_1(f_2(...(f_D(x))))$, the reasoning depth is $D$. This metric captures the inherent logical complexity of the problem, independent of how many intermediate steps a model uses to express the solution.

**Curriculum Learning.** Through gradually increased task difficulty along both reasoning and visual dimensions, our curriculum consists of three levels (§B.2) across five fine-grained tiers (§B.3).

## 3.3 SUPERVISED FINETUNING WITH VISUAL GROUNDED LEARNING

Our training methodology employs supervised fine-tuning with a multi-objective learning framework that combines reasoning generation, visual grounding, and mask-based spatial understanding.

**Training Objective.** The overall training loss combines multiple components, where $\mathcal{L}_{\text{LM}}$ is the standard language modeling loss and $\mathcal{L}_{\text{mask}}$ is our novel mask-based grounding loss with weight $\alpha$:

$$\mathcal{L} = (1 - \alpha)\mathcal{L}_{\text{LM}} + \alpha\mathcal{L}_{\text{mask}} \quad (4)$$

**Language Modeling Loss.** Reasoning generation follows standard auto-regressive training, where $r_{t,i}$ represents the $i$-th token in the $t$-th reasoning step $R_t$, and $r_{<t,i}$ denotes all previous tokens:

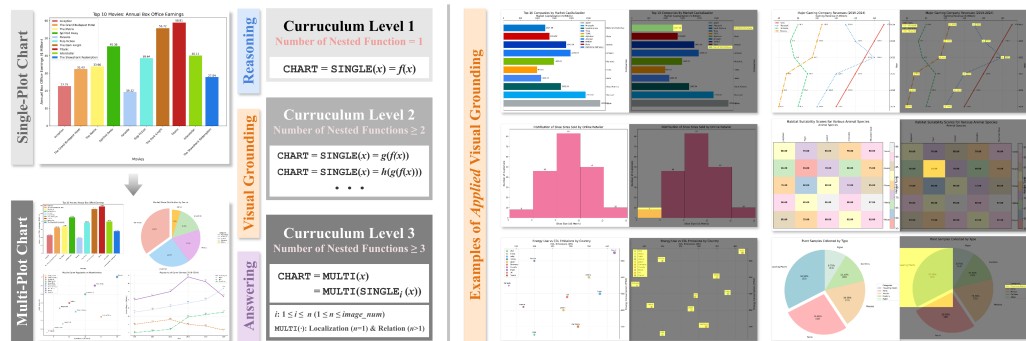

Figure 5: **Multi-Level Curriculum Construction.** We construct CCQA through reasoning decomposition, interleaving visual reasoning, and reasoning chain composition. Examples of **applied** visual grounding is shown on the right. More examples can be found in § C.

$$\mathcal{L}_{\text{LM}} = -\sum_{t=1}^{T} \sum_{i=1}^{|R_t|} \log p(r_{t,i}|\mathcal{I}, Q, r_{<t,i}, \theta) \tag{5}$$

**Visual Grounding Loss.** For spatial understanding, we introduce a mask-based grounding loss (Eq. 6) that directly optimizes the model's ability to ground reasoning steps in visual regions. Here, $\mathcal{L}_{\text{BCE}}$ and $\mathcal{L}_{\text{Dice}}$ are *focal binary cross-entropy loss* (Eq. 24) and *Dice loss* (Eq. 26) respectively, with weights $\beta$ and $\gamma$:

$$\mathcal{L}_{\text{mask}} = \beta \mathcal{L}_{\text{BCE}} + \gamma \mathcal{L}_{\text{Dice}} \tag{6}$$

# 4 CCQA: CURRICULUM CHART QUESTION ANSWERING

## 4.1 DATASET CONSTRUCTION PRINCIPLES

Supporting 🌸 CURV (§3), we introduce *Curriculum Chart Question Answering* (**CCQA**), a systematically constructed curriculum learning dataset that progressively develops visual reasoning capabilities through structured difficulty levels. Our dataset construction follows three core principles:

**Progressive Complexity.** We implement three-level curriculum through systematic variation in reasoning depth ($D$), chart complexity, and operation sophistication (Fig. 5).

**Interleaved Visual Grounding.** Each reasoning step $R_t$ is paired with corresponding ground-truth bounding boxes $B_t^*$ and binary masks $M_t^*$, enabling direct alignment of visual grounding.

**Template-Based Accuracy.** We employ synthetic templates (§B.1) to ensure data accuracy and systematic coverage of reasoning patterns, effectively supporting a stable progression in curriculum learning. As shown in the example below, all chart-specific features are replaced by plotting data:

> **QUESTION:** What is the $<$ y_axis_title $>$ of the $<$ object_singular $>$?

## 4.2 DATASET CONSTRUCTION

**Chart Types.** We leverage seven chart types to endow our dataset with high visual diversity (Fig. 6 & Tab. 4): *bar*, *histogram*, *scatter*, *line*, *heatmap*, *pie*, and *radar*.

**Data Category.** We define 30 domain categories (Tab. 4), employing GPT-4o (OpenAI, 2024) to generate plotting data for chart drawing (Fig. 19).

**QA Types.** Our curriculum templates (§B.2) cover various operations (Tab. 5) on chart components to subplots.

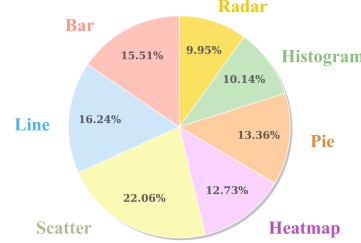

Figure 6: **Distribution of Chart Types.**

**Curriculum with Meta Learning.** Our dataset construction implements a meta-learning paradigm that maximizes visual reasoning generalization while minimizing visual overfitting (§B.4). With only 30 unique charts for each chart type (*i.e.,* 7 types × 30 categories = 210 base images), hundreds

of diverse 'query-reasoning-grounding-answer' quadruplets are derived from each image through systematic template instantiation (§B.2). Our approach aims to guide MLLMs to learn to reason with accurate visual grounding, rather than memorizing specific visual appearances. Therefore, for each base chart image $\mathcal{I}_j$, multiple CQA instances $\{(Q_k, \{D_d, B_d^*\}_{d=1}^D, A_k)\}_{k=1}^K$ where $K \gg 1$, are generated to help the model develop robust visual reasoning capabilities transferable across diverse chart appearances, data distributions, contexts and domains, as well as task complexities.

**Generalize To Real-World Charts And Domains.** Inspired by VDLM (Wang et al., 2025d) that bridges the gap between low-level perception and high-level reasoning, we construct CCQA to underline the significance of enhancing model's understanding of basic visual components and spatial features for accurate visual reasoning (§6.4). CURV finetuned on CCQA are also applicable to real-world chart understanding and out-of-domain benchmarks (§5.2), validating not only the effectiveness but also the adaptability and generalizability of our approach (§A.1 & B.4).

# 5 EXPERIMENTS

## 5.1 SETUP

**Baseline.** We use two close-source models, GPT-4o (OpenAI, 2024) and GPT-4.1-mini (OpenAI, 2025), and seven open-source MLLMs, Llama-3.2-Vision (AI, 2024), Gemma-3 (Google, 2024), InternVL3 (OpenGVLab, 2025), and Qwen2.5-VL (Qwen, 2025) with different model sizes, as comparison baselines.

**Model.** We employ five MLLMs as our base model finetuning for multi-turn chart reasoning with visual grounding: Qwen2.5-VL (3B and 7B) and InternVL-3 (1B, 2B and 8B). Respectively, we consider three variants of CURV (Tab. 2) through different grounding strategies (§C).

**Data.** We split CCQA into *training* and *test* sets. All models are evaluated on the *test* set unseen for finetuned models. In particular, the base model is finetuned on levels 1-2 of CCQA (§4), where task complexity is progressively increased. Same as baselines, finetuned models are evaluated on the *test* sets of curriculum levels 1-3 to examine their in-domain performance as well as their robustness and generalizability to more challenging tasks. Implementation details can be found in §F.1.

## 5.2 EVALUATION

**Evaluation Data.** We evaluate model on CCQA (§4) and chart and out-of-domain benchmarks:

- **CCQA:** Highlighting the significance of learning chart basics through increased task difficulty, we evaluate models on the test sets of CCQA covering three increased curriculum levels, respectively. CQA samples with labeled charts are randomly select to construct three-level test sets.
- **Chart Benchmarks:** Adapting to challenging chart understanding, we additionally evaluate on four popular CQA benchmarks, including *ChartMuseum* (Tang et al., 2025), *CharXiv* (Wang et al., 2024), *ChartQA* (Masry et al., 2022), and *ChartQAPro* (Masry et al., 2025).
- **Out-of-Domain Benchmarks:** Generalizing to different domains of multimodal reasoning, we extend our evaluation to multi-discipline multimodal reasoning tasks, including MathVista (Lu et al., 2023) and MMMU-Pro (Yue et al., 2025).

**Evaluation Metrics.** We evaluate different aspects of responses through complementary metrics that take both textual outputs and visual grounding into consideration (§E):

(1) *Answer*: We define CQA accuracy as the mean *answer* accuracy across all testing CQA instances. Specifically, we employ three accuracy metrics:

**LLM As Judge ($acc@LLM$):** We employ GPT-4.1-mini (OpenAI, 2025) as the judge to evaluate answer accuracy through *True-or-False* assessment ($pass@1$). More details are discussed in §E.1.

**Rule As Judge ($acc@range$):** To mitigate potential biases introduced by LLM-as-judge (Dorner et al., 2024; Li et al., 2024a), we introduce a rule-based evaluation metric (Algorithm 1) that assesses answer accuracy through systematic parsing and rule-based judgment. In particular, it incorporates four *range* criteria that capture different levels of strictness (§E.1), including the absolute accuracy ($acc@0.0$) and three progressively relaxed thresholds ($acc@0.05$, $acc@0.1$, $acc@0.2$).

Table 2: **Performance On CCQA.** Performance of various models and baselines across different curriculum levels using five accuracy evaluation metrics $acc@X$ where $X$ is LLM or ranges (§ 5.2).

| Model | Size | Level 1 | | | | | Level 2 | | | | | Level 3 | | | | |
|---|---|---|---|---|---|---|---|---|---|---|---|---|---|---|---|---|
| | | $@LLM$ | @0.0 | @0.05 | @0.1 | @0.2 | $@LLM$ | @0.0 | @0.05 | @0.1 | @0.2 | $@LLM$ | @0.0 | @0.05 | @0.1 | @0.2 |
| *Close-Source MLLMs* | | | | | | | | | | | | | | | | |
| GPT-4o | - | 57.64 | 54.07 | 62.00 | 65.36 | 70.50 | 34.04 | 33.75 | 44.93 | 50.54 | 57.96 | 22.14 | 22.29 | 30.25 | 34.04 | 39.14 |
| GPT-4.1-mini | - | 70.86 | 67.43 | 76.29 | 78.79 | 79.93 | 37.61 | 36.54 | 47.18 | 52.93 | 60.86 | 26.14 | 25.46 | 33.11 | 37.79 | 42.32 |
| *Open-Source Baselines* | | | | | | | | | | | | | | | | |
| Gemma-3 | 4B | 38.21 | 28.64 | 32.79 | 37.00 | 41.29 | 18.07 | 12.86 | 18.82 | 23.64 | 28.29 | 11.43 | 9.64 | 13.14 | 15.54 | 18.96 |
| Llama-3.2-V | 11B | 44.86 | 38.29 | 40.86 | 42.86 | 47.07 | 23.43 | 18.25 | 22.46 | 25.29 | 29.39 | 16.57 | 14.25 | 16.54 | 18.29 | 20.29 |
| InternVL3 | 1B | 20.54 | 16.38 | 20.53 | 24.40 | 29.68 | 8.51 | 7.44 | 12.61 | 15.37 | 20.65 | 6.76 | 5.87 | 8.31 | 10.09 | 10.99 |
| | 2B | 33.53 | 32.52 | 37.52 | 42.52 | 47.29 | 13.11 | 13.22 | 18.85 | 27.14 | 35.55 | 10.69 | 11.18 | 14.71 | 18.72 | 24.90 |
| | 8B | 46.79 | 44.29 | 52.14 | 57.71 | 63.00 | 25.75 | 25.57 | 35.36 | 41.64 | 50.79 | 17.84 | 17.48 | 24.57 | 28.93 | 34.68 |
| Qwen2.5-VL | 3B | 45.25 | 43.52 | 51.54 | 56.25 | 61.54 | 22.75 | 22.86 | 31.14 | 37.21 | 45.36 | 16.18 | 16.00 | 21.89 | 25.82 | 31.71 |
| | 7B | 54.21 | 50.79 | 60.43 | 64.64 | 69.29 | 28.68 | 28.82 | 39.93 | 45.54 | 52.61 | 19.01 | 19.38 | 26.71 | 31.50 | 36.93 |
| *Ours* | | | | | | | | | | | | | | | | |
| **Applied** (InternVL3) | 1B | 25.79 | 21.36 | 25.29 | 31.43 | 37.86 | 10.57 | 9.25 | 12.89 | 17.18 | 22.21 | 7.11 | 7.36 | 9.25 | 11.00 | 12.93 |
| | 2B | 42.64 | 41.79 | 48.71 | 55.36 | 62.14 | 18.68 | 19.29 | 25.79 | 32.93 | 41.50 | 11.39 | 12.57 | 16.61 | 20.75 | 26.32 |
| | 8B | 58.86 | 54.64 | 66.29 | 69.14 | 71.79 | 34.47 | 33.90 | 49.67 | 56.47 | 63.91 | 18.87 | 18.71 | 27.99 | 31.97 | 37.41 |
| **Applied** (Qwen2.5-VL) | 3B | 54.21 | 51.21 | 59.50 | 65.00 | 68.50 | 25.86 | 27.14 | 39.18 | 47.25 | 55.18 | 16.86 | 17.32 | 23.93 | 28.61 | 33.11 |
| | 7B | 65.79 | 59.14 | 71.93 | 75.29 | 78.29 | 36.82 | 34.79 | 50.82 | 56.18 | 62.75 | 21.04 | 21.11 | 30.21 | 34.25 | 39.39 |
| **Boxed** (Qwen2.5-VL) | 3B | 58.07 | 51.79 | 61.00 | 65.71 | 69.86 | 25.96 | 25.32 | 37.75 | 45.89 | 53.86 | 16.92 | 16.82 | 22.89 | 27.21 | 33.29 |
| | 7B | 59.79 | 52.79 | 71.93 | 74.57 | 76.64 | 33.79 | 30.32 | 49.75 | 55.89 | 62.21 | 20.14 | 18.04 | 28.89 | 32.32 | 36.29 |
| **Cropped** (Qwen2.5-VL) | 3B | 49.71 | 46.79 | 54.86 | 60.07 | 65.21 | 20.68 | 21.11 | 32.61 | 40.36 | 50.82 | 15.00 | 18.89 | 20.96 | 24.71 | 30.25 |
| | 7B | 58.93 | 56.57 | 67.71 | 72.93 | 76.71 | 30.04 | 29.71 | 40.43 | 46.71 | 53.79 | 18.68 | 18.71 | 26.79 | 30.36 | 35.11 |

**(2)** *Reasoning*: For reasoning evaluation, we employ two complementary approaches:

**Micro Evaluation** ($acc@mic$): We evaluate reasoning steps using a combination of five micro metrics: ROUGE-L (Eq. 17), BLEU (Eq. 18), METEOR (Eq. 20), BERTSCORE (Eq. 19), and COSINE SIMILARITY (Eq. 21).

**Macro Evaluation** ($acc@mac$): We employ GPT-4.1-mini as the judge to implement macro evaluation by assigning '0-10' quality scores to model reasoning chains based on three criteria (§E.2).

**(3)** *Visual Grounding*: For visual assessment, we leverage Intersection-over-Union (IoU) variants CIOU (Eq. 22) and GIOU (Eq. 23), where GIOU (Rezatofighi et al., 2019) is generalized IoUs and CIOU evaluates the cumulative intersection over the cumulative unions (Zheng et al., 2019).

# 6 RESULTS

## 6.1 MAIN RESULTS

Equipping MLLMs with visual grounded reasoning, our evaluation is based on a variety of datasets and domains (§5.2), extending from our multi-level curriculum datasets to chart understanding benchmarks and out-of-domain multimodal reasoning.

Table 3: **Performance on Chart and Out-of-Domain Benchmarks.** LLM-as-judge $acc@LLM$ (§ 5.2) on chart understanding and out-of-domain benchmarks.

| Model | Size | Chart Benchmarks | | | | Out-of-Domain | |
|---|---|---|---|---|---|---|---|
| | | ChartQA | ChartQA-Pro | CharXiv | ChartMuseum | MathVista | MMMU-Pro |
| *Baselines* | | | | | | | |
| InternVL3 | 1B | 41.68 | 11.48 | 15.70 | 10.01 | 35.80 | 9.94 |
| | 2B | 66.96 | 20.84 | 24.30 | 15.02 | 55.90 | 16.36 |
| | 8B | 74.56 | 30.56 | 36.20 | 24.42 | 68.80 | 26.99 |
| Qwen2.5-VL | 3B | 62.32 | 17.02 | 19.50 | 12.21 | 56.10 | 21.45 |
| | 7B | 72.48 | 29.77 | 32.50 | 21.62 | 64.60 | 28.21 |
| *Ours* | | | | | | | |
| **Applied** (InternVL3) | 1B | 50.52 | 14.17 | 17.70 | 8.11 | 41.48 | 11.68 |
| | 2B | 68.24 | 22.41 | 26.10 | 13.71 | 58.52 | 17.24 |
| | 8B | 77.36 | 31.37 | 37.70 | 21.12 | 70.50 | 28.73 |
| **Applied** (Qwen2.5-VL) | 3B | 71.48 | 23.87 | 30.70 | 17.82 | 60.93 | 21.83 |
| | 7B | 74.36 | 31.72 | 33.80 | 23.02 | 66.81 | 29.61 |

**Performance On Levels 1-2 of CCQA.** Comparing with baselines (Tab. 2), our finetuned models achieves consistently higher accuracy across all metrics (*test set*), spanning from LLM-based evaluation to rule-based assessment ($acc@ \pm range$). The highest performance is achieved by CURV@Applied(Qwen2.5-VL-7B), showcasing up to 10.79% absolute gains in comparison to its base model (Qwen2.5-VL-7B) on LLM-based judgment, and up to 6.50% accuracy improvements on strict rule-based evaluation ($acc@0.0$). Compared to GPT-4o, it also presents 7.36% and 2.78% higher LLM-judged accuracy on levels 1 and 2, respectively, demonstrating the effectiveness of CURV in endowing the model with enhanced visual reasoning abilities.

**Performance On Complex Chart Understanding.** Although trained solely on single-plot chart understanding ($1 \leq D \leq 3$), finetuned models exhibit generalizabilities to multi-plot charts with $D \geq 3$. As shown in Tab. 2, CURV@Applied (Qwen2.5-VL-7B) achieves up to 3.50% accuracy gain, and CURV@Applied (Qwen2.5-VL-3B) also shows 2.79% improvement across all metrics.

**Performance On Chart Benchmarks.** Aiming for ✿CURV to be not only adaptable across different task complexities but also generalizable to real-world chart comprehension, we extend our evaluation to CQA benchmarks (§5.2). Results in Tab. 3 highlight the strong generalizability of ✿CURV, with improvements of more than 1.30% across four chart benchmarks.

**Generalizability To Out-of-Domain Multimodal Reasoning.** As shown in Tab. 3, the performance advantage of ✿CURV remains consistent in out-of-domain multimodal reasoning across diverse categories (Yue et al., 2025; Lu et al., 2023), attaining up to 4.83% accuracy improvements.

## 6.2 INVESTIGATION ON DIFFERENT VISUAL GROUNDING APPROACHES

Implementing three distinct visual grounding methods (§3 & §C), results in Tab. 2 unveils that visual reasoning enhancement effects of **boxed** grounding stay less beneficial than explicitly highlighting the regions of focus through **applied** masking, despite its simplicity and straightforwardness. On the other hand, although restricted by the trade-off between zoom-in resolution and computation overhead (§F.2), **cropped** grounding unveils its strengths in chart reasoning (Tab. 2), yielding up to a $\uparrow 4.72\%$ improvement in $acc@LLM$ and a $\uparrow 5.78\%$ improvement in $acc@0.0$ despite a sixteen-fold reduction in resolution.

## 6.3 EFFECTS OF CURRICULUM LEVELS & CHART TYPES

**Effects of Chart Types.** We compare the accuracy improvements ($\Delta_{acc}$) of ✿CURV@*Applied* (Qwen2.5-VL-7B) against its base model (Fig. 7). Bar charts contribute the most for accuracy enhancement, followed by line plots and heatmaps. Other types of charts also demonstrate positive effects on CQA performance, with radar charts contributing the least, possibly because they are less commonly used.

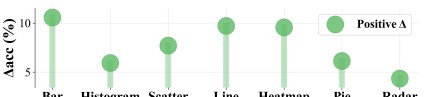

Figure 7: **Effects of Chart Types.** We compare $\Delta_{acc}$ between Qwen2.5-VL-7B and its *applied* versions trained on seven chart types, respectively.

**Effects of Curriculum Levels.** Fig. 8 illustrates the effects of different curriculum levels. Although all three curriculum levels exhibit positive improvements on overall accuracy, levels 1+2 earns the greatest benefit among all. Comparing among training on levels 1, 1+2, and 3, the notable improvements of levels 1 and 1+2 validate the significance of foundational learning in effective chart understanding across varying difficulty levels (§6.4).

## 6.4 SIGNIFICANCE OF FOUNDATIONAL LEARNING

Fig. 8 reveals a critical finding that strongly validates our curriculum learning design (§4). While training on level 1 alone provides solid foundational performance ($\uparrow 10.79\%$ on level 1), progressive training on levels 1+2 demonstrates the optimal learning accumulation, achieving the best overall performance across all difficulty levels ($\uparrow 11.58\%$ on level 1, $\uparrow 8.14\%$ on level 2, $\uparrow 2.03\%$ on level 3). However, extending training to include level 3 unfolds a concerning trade-off between robustness and generalizablility: While notably improves complex rea-

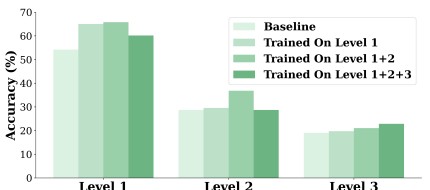

Figure 8: **Foundational Learning In Chart Understanding.** We compare Qwen2.5-VL-7B with its *applied* model trained on curriculum levels $1$, $1+2$, and $1 + 2 + 3$, respectively.

soning performance ($\uparrow 3.81\%$ on level 3), it significantly degrades foundational reasoning abilities, only higher than baseline by $\uparrow 5.93\%$ and $\uparrow 0.03\%$ on levels 1 and 2, respectively.

## 6.5 CHALLENGES IN MULTI-CHART UNDERSTANDING

Curriculum level 3 introduces notably increased complexity that challenges even advanced MLLMs (Fig. 9). Compared to level 1 (20.21-70.86% accuracy range) and level 2 (8.51-37.61% accuracy range), level 3 ($\leq 26.14\%$) presents a substantial complexity gap. The transition from single-chart reasoning in levels 1-2 to multi-chart scenarios in level 3 fundamentally amplifies both logical reasoning and visual comprehension demands, as models need to simultaneously process multiple visual contexts while maintaining coherent reasoning chains across disparate chart elements. The op-

erational breakdown further reveals that **localization** tasks pose significant challenges with accuracy below 35% for most models. Moreover, **relation** interpretation that involve cross-chart reasoning between subplots achieve even lower performance, highlighting the limitations in current MLLMs' capacity for multi-plot cross-context visual reasoning.

Taken together, these findings illuminate how multi-plot chart understanding presents a qualitative leap in complexity, instead of a smooth extension of earlier levels. The steep drop in performance, combined with the compounded challenges of localization and relational reasoning, suggests that effective progress in chart understanding depends first on consolidating the foundational competencies established in foundational single-chart scenarios before advancing to multi-plot reasoning (§6.4).

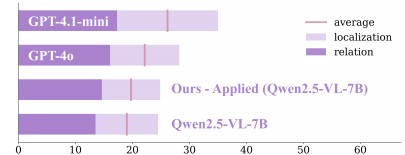

Figure 9: **Challenges In Multi-Chart Understanding.** Visualization of top-4 accuracy scores on curriculum level 3.

## 7 RELATED WORK

**Chain-of-Thought Reasoning.** CoT reasoning has emerged as foundations for enhancing the interpretability and performance of large language models (LLMs). Even prompting the LLMs to do CoT reasoning before answering can lead to improved performance (Wei et al., 2022; Wang et al., 2023). CoT reasoning is particularly beneficial for MLLMs in complex visual reasoning tasks where attentions are interrelated to both visual and textual features (Zhang et al., 2024b; Shao et al., 2024; Zhao et al., 2025; Qi et al., 2024). Recent work focuses on improving CoT reasoning abilities through various approaches. (Zhang et al., 2024b) propose a two-fold approach that first distills rationales from GPT-4o to enrich training data, then applies reinforcement learning to calibrate reasoning quality. Similarly, (Chen et al., 2024) introduce a two-stage training framework that employs supervised finetuning on step-by-step reasoning samples, followed by LLM feedback incorporation to produce highly consistent and grounded reasoning chains. The integration of visual manipulations in reasoning processes also shows promising gains (Qi et al., 2024) by enabling MLLMs to solve problems step-by-step.

**Multimodal Chart Understanding.** CQA represents a specialized domain that requires accurate understanding of structured visual data representations and complex reasoning over visual and textual chart elements in addition to the language inputs. Recent datasets pay attention to real-world chart complexity and diversity. In addition to ChartQA (Masry et al., 2022) involving visual and logical reasoning over charts, ChartMuseum (Tang et al., 2025) is introduced with substantial performance gaps between models and humans. Aligning to scientific research, CharXiv (Wang et al., 2024) presents a comprehensive evaluation suite with more than 2,000 challenging charts extracted from arXiv papers. Supported by chart benchmarks, SIMPLOT (Kim et al., 2024) proposes a two-step method to extract elements necessary for chart reasoning. Paying more attention to visual reasoing, the Graph-of-Thought (GoT) guided compositional reasoning model (Zhang et al., 2024a) is introduced for multi-step reasoning through directed acyclic GoT. In addition to the work that advances CQA through chart component recognition (Zheng et al., 2024), (Li et al., 2024b) addresses the reasoning challenges in CQA by leveraging LLMs to generate synthetic question-answer pairs.

## 8 CONCLUSION

We present ✿CURV, a curriculum learning framework that develops intrinsic visual reasoning capabilities through progressive multi-step visually grounded reasoning. To better support model learning, we systematically construct CCQA with three progressive difficulty levels. Results demonstrate that tightly interleaving reasoning with visual grounding throughout training enables models to achieve consistent performance improvements across curriculum levels and generalize effectively to real-world chart understanding and out-of-domain multimodal reasoning. Our work establishes a foundation for developing self-contained visual reasoning capabilities in MLLMs, moving beyond extrinsic assistance toward intrinsic grounded visual reasoning. Building on our model-level enhancement, for future work, we aim to explore agentic chart understanding to better incorporate external knowledge, tools, and collaboration, to complement intrinsic visual reasoning of individual MLLMs.

USE OF LLMs

We used LLMs (e.g., ChatGPT) to assist with grammar correction. In a few cases, we also used LLMs to improve the conciseness of overly long sentences.

REPRODUCIBILITY STATEMENT

We will make the complete source code and curriculum learning datasets public to ensure reproducibility of our work. In this paper, we also elaborate our implementation details, hyperparameter settings, and prompts for LLM-as-judge evaluation to assist the reproduction of our work.

ETHICS STATEMENT

In this work, we introduce a curriculum learning framework with the dataset CCQA constructed through meta-learning supported CQA creation. Other evaluation benchmarks, including chart understanding and multimodal reasoning, are publicly available and do not contain personally identifiable information or sensitive content. Our methods are designed for research and educational purposes, and we do not foresee direct misuse. With every step being effectively controlled, we positively believe that our work does not violate any ethical standards.

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

CONTENTS

## A PRELIMINARY EXPLORATION & VALIDATION

### A.1 PRELIMINARY EXPLORATION ON MOTIVATIONS

Building on the cognitive perspectives that humans solve multimodal problems through **decomposition**, **interleaved visual reasoning**, and **composition** (Fig. 1), we conduct preliminary studies to examine where current MLLMs fall short (§2).

Concretely, we evaluate GPT-4o on 60 CharXiv samples, categorizing the root causes of CQA failures into three classes: *reasoning errors* (**decomposition**), *perception errors* (**interleaved visual reasoning**), and *integration errors* (**composition**). As shown in Fig. 3, perception errors (28.33%) emerge as the dominant reason of failures, reflecting the difficulty of accurately grounding reasoning steps in fine-grained chart details. Beyond perception, models also exhibit weaknesses in decomposition (10.00%), struggling to break down complex problems into coherent chains of reasoning. Additionally, they also show deficiencies in composition (8.33%), failing to integrate grounded visual evidence into a coherent, interleaved chain of thought. These findings reveal systematic shortcomings in human-inspired reasoning stages, motivating our design of ⚙️CURV that enhances MLLMs' intrinsic visual grounded reasoning capabilities by simulating human cognitive process of **decomposing**, **interleaving**, and **composing** toward a solution.

Inspired by recent findings that models can effectively learn from low-level features (Wang et al., 2025d), we concretize the notion of **decomposition** in two complementary forms: (1) *visual decomposition*, where each chart is decomposed into low-level components (Fig. 10) to to guide MLLMs' attention toward fine-grained and informative details; and (2) *reasoning decomposition*, where each chart understanding problem is decomposed into a structured chain of reasoning steps to help MLLMs enhance their logical reasoning capacities. Building on these decompositions, we incorporate **interleaving** insights into our design, enabling reasoning to be interleaved with dynamically shifting visual focuses. Accordingly, the **composition** process integrates all intermediate learning in a coherent chain: from *reasoning composition* that consolidates step-wise reasoning into a coherent logical chain, to *visual composition* that progressively aggregates low-level visual interpretations into holistic chart comprehension.

Together, these elements form the foundation of our **curriculum learning** design (§3), which standardizes two dimensions of progression: (1) *curriculum CQA reasoning difficulty*, controlled by increasing the number of nested functions; and (2) *curriculum chart visual complexity*, controlled by increasing the number of low-level components, chart types, and chart subplots. We further support our design with **meta-learning** (§B.4) to endow MLLMs with adaptability and generalizability in the face of varying chart types, context domains, and task complexity.

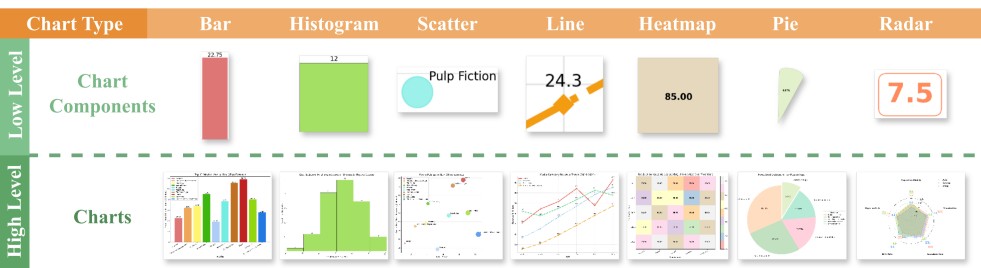

Figure 10: **From Low-Level Chart Components To High Level Charts.** We decompose all kinds of charts into low-level components to endow MLLMs with both foundational chart understanding abilities and adaptive generalizabilities to high-level complexities.

### A.2 PRELIMINARY EXPLORATION ON CQA CHALLENGES

To investigate the underlying causes of failures in chart understanding, we employ five MLLMs, including three *close-source* models (GPT-4.1-mini (OpenAI, 2025), GPT-4o (OpenAI, 2024),

Gemini-2.5-Flask (DeepMind, 2025)) and two *open-source* models (Qwen2.5-VL-3B and Qwen2.5-VL-7B (Qwen, 2025)) to identify the root causes of their failures. Specifically, we analyze their CQA outputs case-by-case on different CQA benchmarks (§5.2), noticing several key patterns in MLLMs' CQA failures:

- **Reasoning Accuracy & Consistency:** While prompting MLLMs to do CoT reasoning can guide them toward correct answers in some cases, we still observe notable visual reasoning failures. For example, in Fig. 11, Qwen2.5-VL-3B misaligns line colors with their corresponding labels at the beginning, which propagates this misperception through subsequent reasoning and results in an incorrect answer. On the other hand, in Fig. 12, GPT-4o fails to excluded "*Loki*" despite having correctly identified it in earlier steps, unveiling the inconsistency in its evolution of reasoning. Another form of reasoning inconsistency emerges in recursive self-correction, where it may occur repeatedly throughout the model's reasoning process, ultimately producing inconsistent or divergent answers (*e.g.*, Gemini-2.5-Flash in Fig. 12).

- **Visual Grounding Accuracy:** MLLMs exhibit significant challenges in precisely capture visual details from the chart images. For example, in Fig. 11, GPT-4o inaccurately estimates the $W_H$ value of the red "fi" point as approximately 0.105, while the true value is significantly less than 0.1, residing just above 0.0.

- **Visual Reasoning Effectiveness:** In complex reasoning tasks requiring the integration of multiple visual regions and reasoning steps, MLLMs often struggle to effectively link visual attention with logical reasoning. For instance (Fig. 12), although GPT-4o and GPT-4o-mini both perceive accurately in their initial perception, they exhibit distinct failures in subsequent reasoning: GPT-4o incorrectly includes "*Roar*" while GPT-4o-mini fails to incorporate "*Loki*".

### A.3 PRELIMINARY VALIDATION ON CCQA

In validating our proposed curriculum learning benchmark, CCQA (§4), we employ the same five MLLMs as our preliminary exploration (§A.2), examining case studies on five fine-grained difficulty tiers (§B.3) of the three curriculum levels (§B.2):

- **Tier 1: Single-Plot Reasoning** ($D = 1$). The example in Fig. 13 is a *Statistics-Mean* query (Tab. 5) that MLLMs often fail to correctly answer. Among all three faiure cases, Qwen2.5-VL-3B and GPT-4o-mini fail at accurately perceive the numbers from the chart image, while Qwen2.5-VL-7B encounters calculation errors despite correct visual understanding.

- **Tier 2: Single-Plot Reasoning** ($D = 2$). The example in Fig. 14 defines a specific *Subset* (Tab. 5) that poses significant obstacles for most MLLMs in arriving at the correct answer. Except for Gemini-2.5-Flask, all other four models struggle to accurately identify the required chart components.

- **Tier 3: Single-Plot Reasoning** ($D \geq 3$). The example in Fig. 15 involves multiple operators (Tab. 5) that unfolds notable challenges for most MLLMs in both visual reasoning and mathematical calculation. Except for GPT-4o, all other four models struggle to accurately calculate the mean, leading to wrong final answers.

- **Tier 4: Multi-Plot Reasoning (Localization).** The example in Fig. 16 involves both *Localization* and *Statistics-Sum* (Tab. 5). All five models fail to correctly perceive the visual information, resulting in wrong intermediate reasoning and incorrect final answer.

- **Tier 5: Multi-Plot Reasoning (Relation).** The example in Fig. 17 additionally pay attention to the *Relation* (Tab. 5) of different subplots. Likewise, the incorrect visual reasoning eventually lead to wrong final answers for all five models.

The diverging trends of failure cases (*increase*) and success cases (*decrease*) (Fig. 13-17) demonstrate the effectiveness of our curriculum design for progressively increased complexity tiers, meanwhile highlighting the challenges that persist in chart understanding across all reasoning depths and task complexity levels.

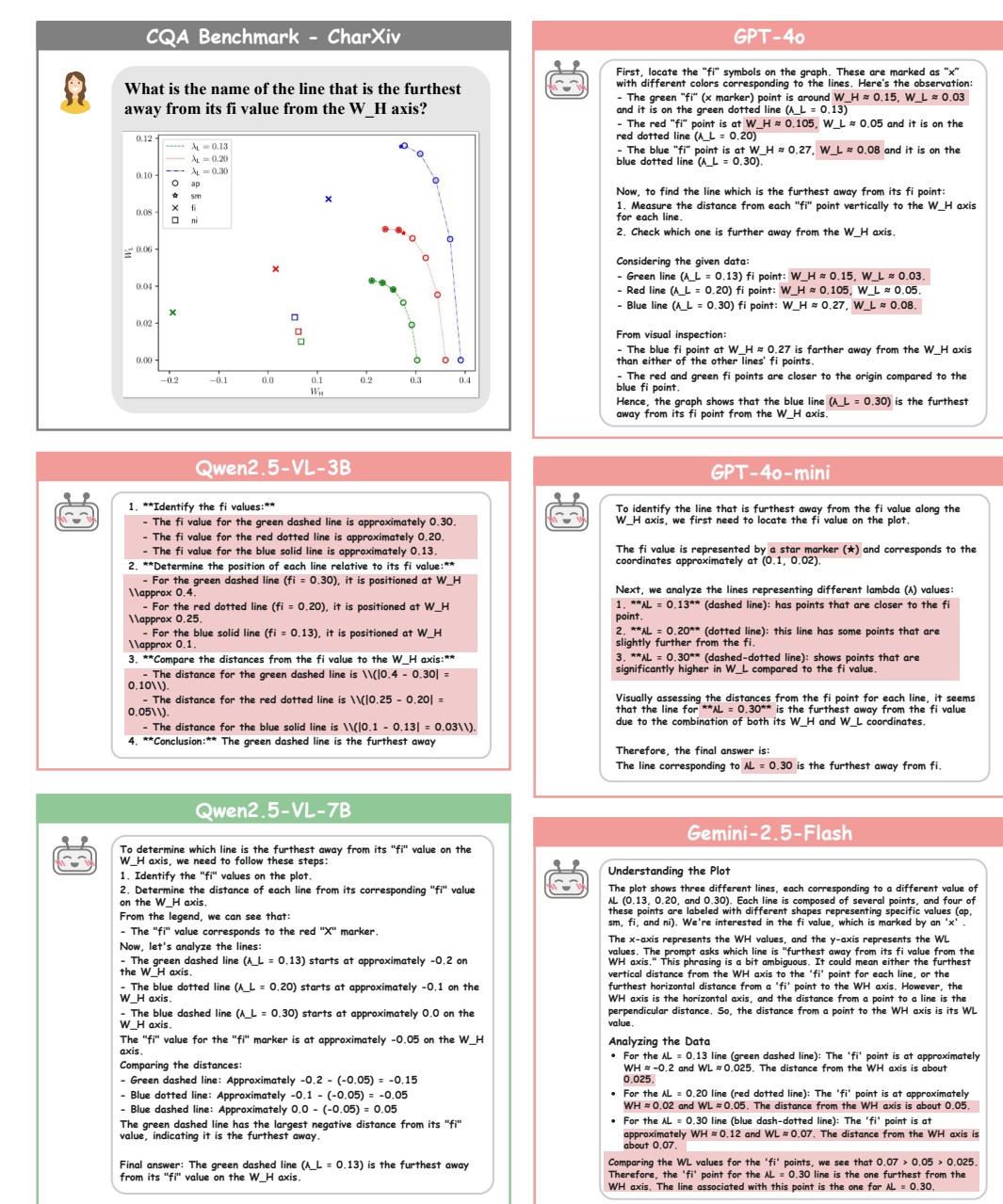

Figure 11: **Examples of Preliminary Exploration on CharXiv.** We conduct case study on CharXiv samples by prompting the model to generate CoT reasoning and final answer. CoT reasoning prompt is omitted in the input for clarify.

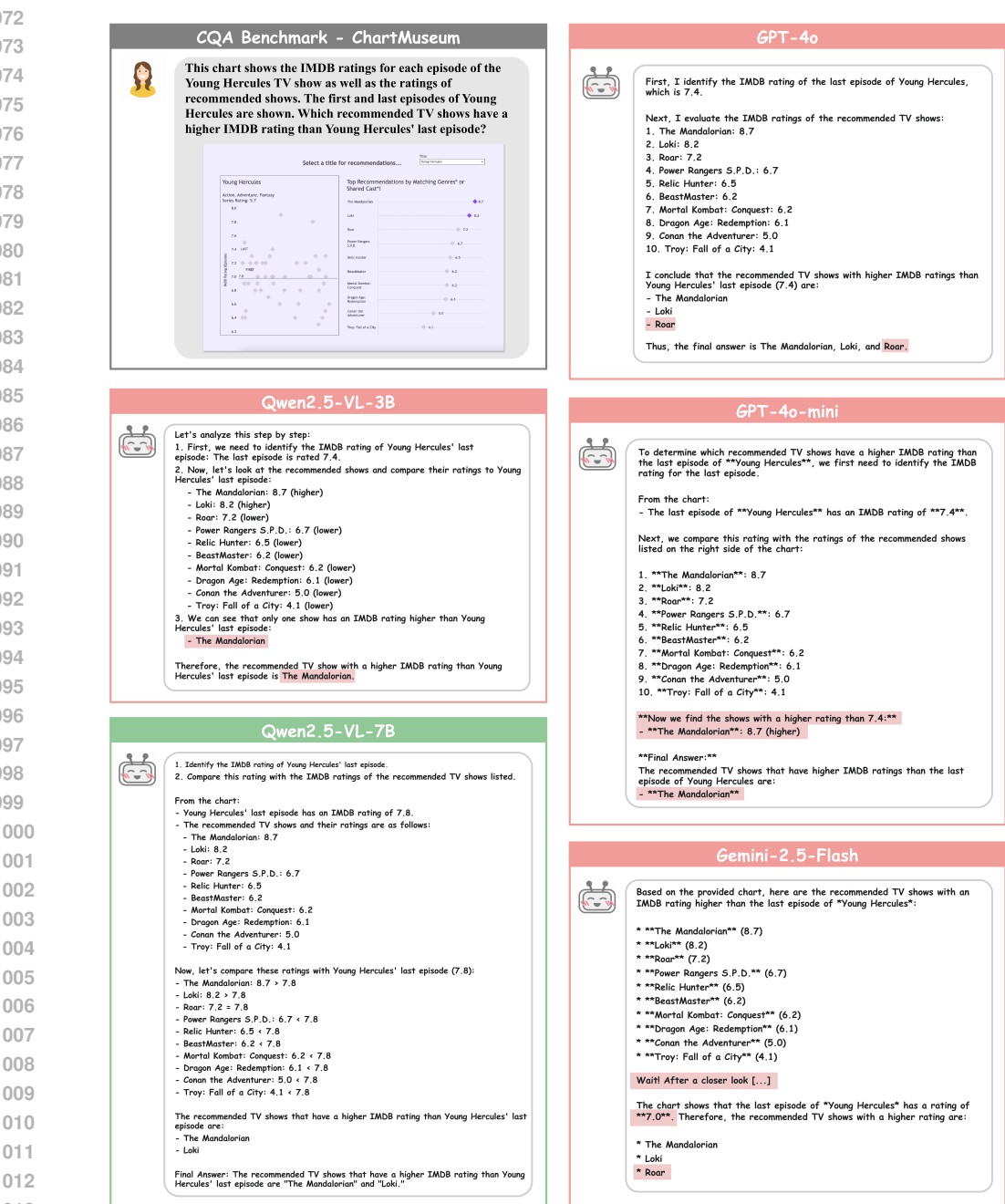

Figure 12: **Examples of Preliminary Exploration on CharXiv.** We conduct case study on CharXiv samples by prompting the model to generate CoT reasoning and final answer. CoT reasoning prompt is omitted in the input for clarify.

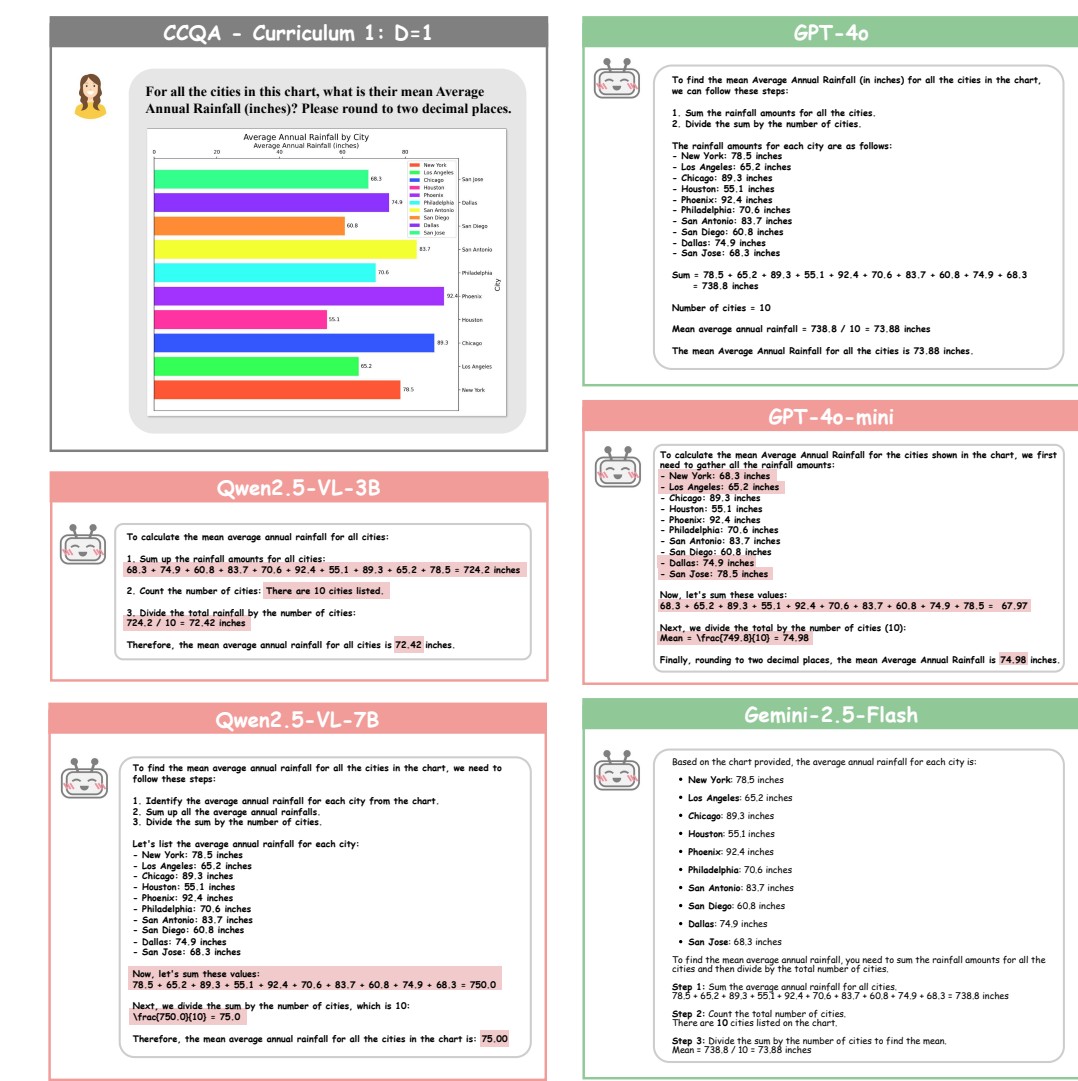

Figure 13: **Examples of Validation On Curriculum Level 1** ($D = 1$). We validate our CCQA on *Tier 1* (§ B.3), i.e., *Curriculum Level 1* ($D = 1$), by prompting the model to generate CoT reasoning and final answer. CoT reasoning prompt is omitted in the input for clarify.

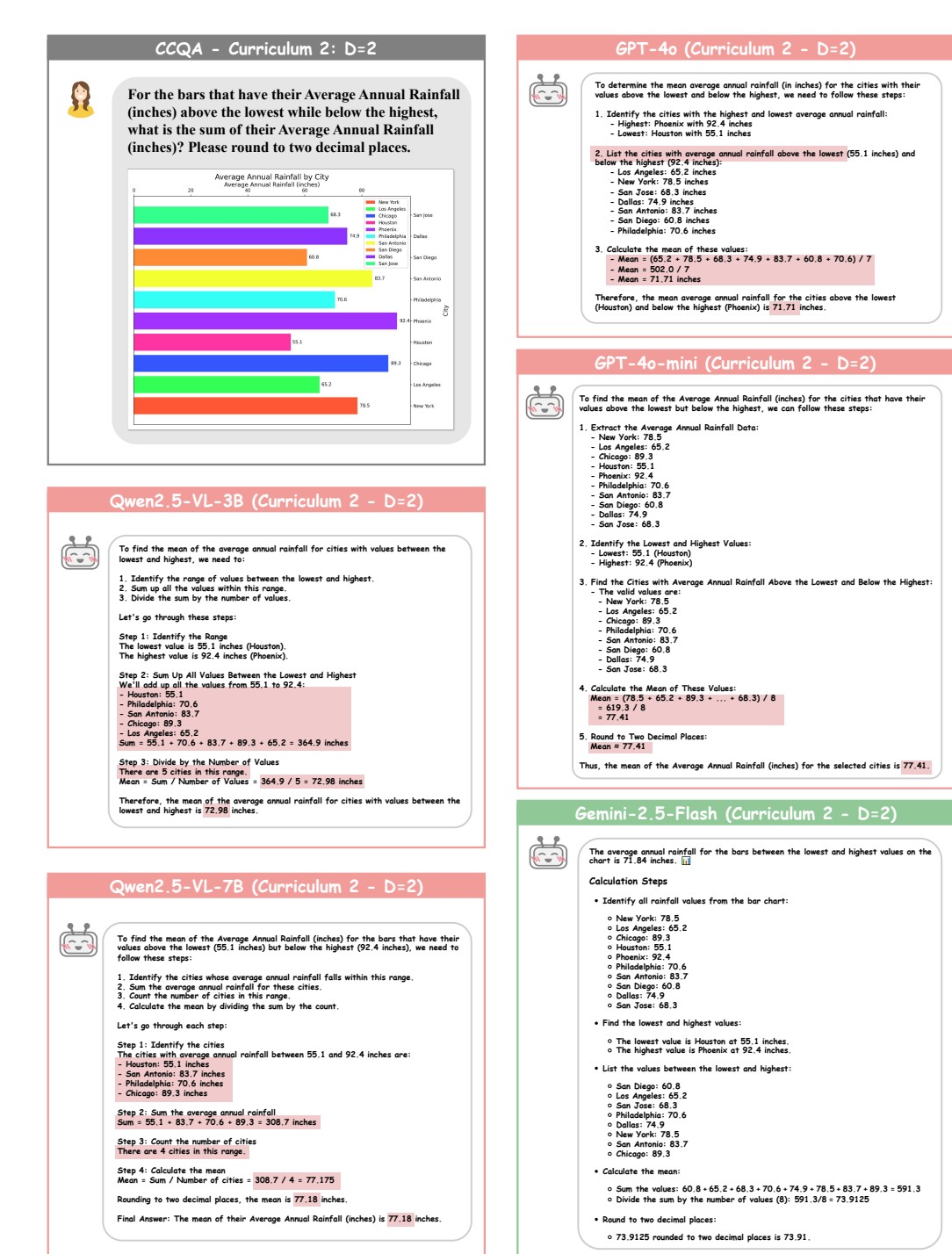

Figure 14: **Examples of Validation On Curriculum Level 2** ($D \geq 2$). We validate our CCQA on *Tier 2* (§ B.3), i.e., *Curriculum Level 2* ($D \geq 2$), by prompting the model to generate CoT reasoning and final answer. CoT reasoning prompt is omitted in the input for clarify.

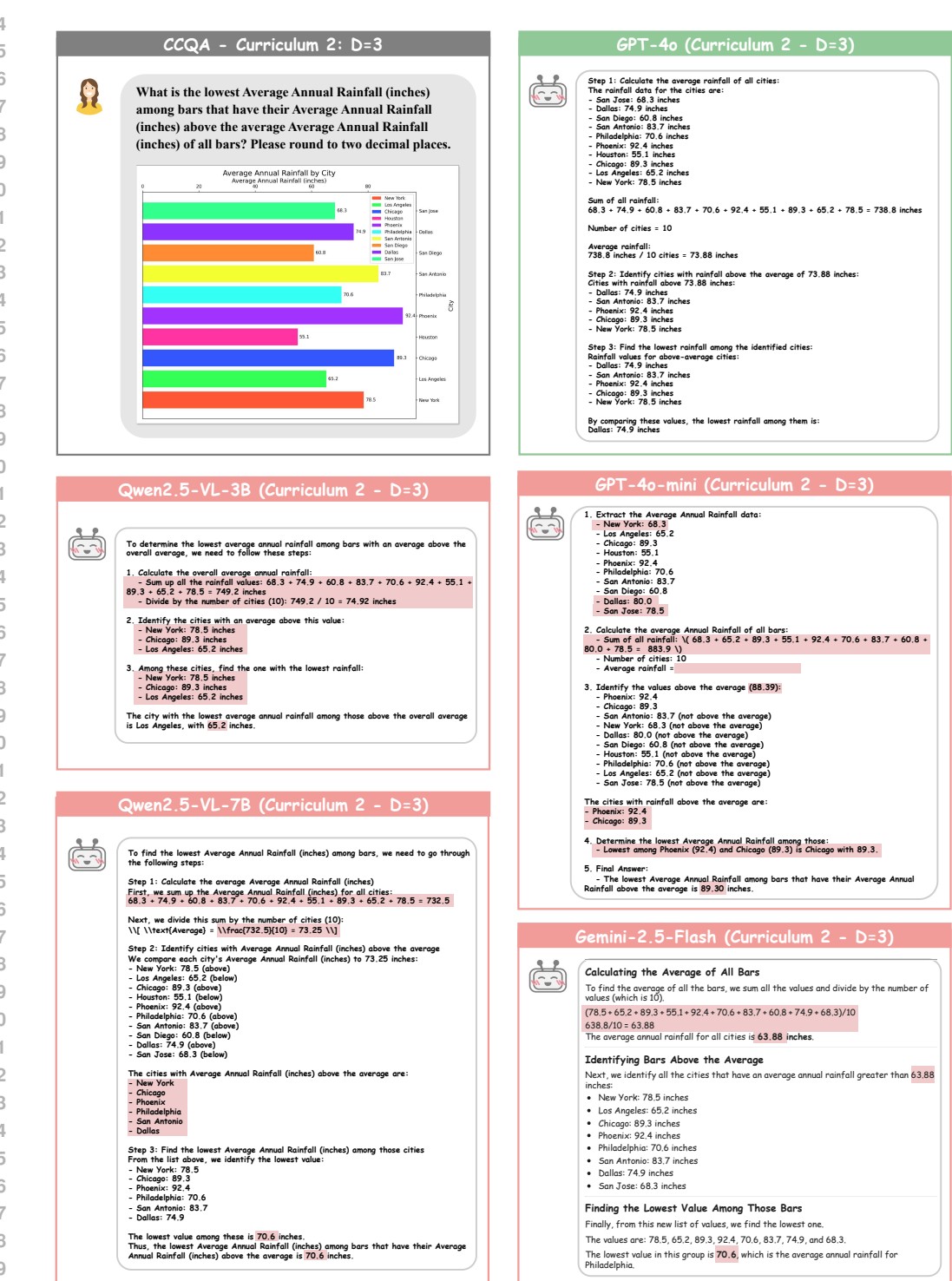

Figure 15: **Examples of Validation On Curriculum Level 2** ($D \geq 3$). We validate our CCQA on *Tier 3* (§ B.3), i.e., *Curriculum Level 2* ($D \geq 3$), by prompting the model to generate CoT reasoning and final answer. CoT reasoning prompt is omitted in the input for clarify.

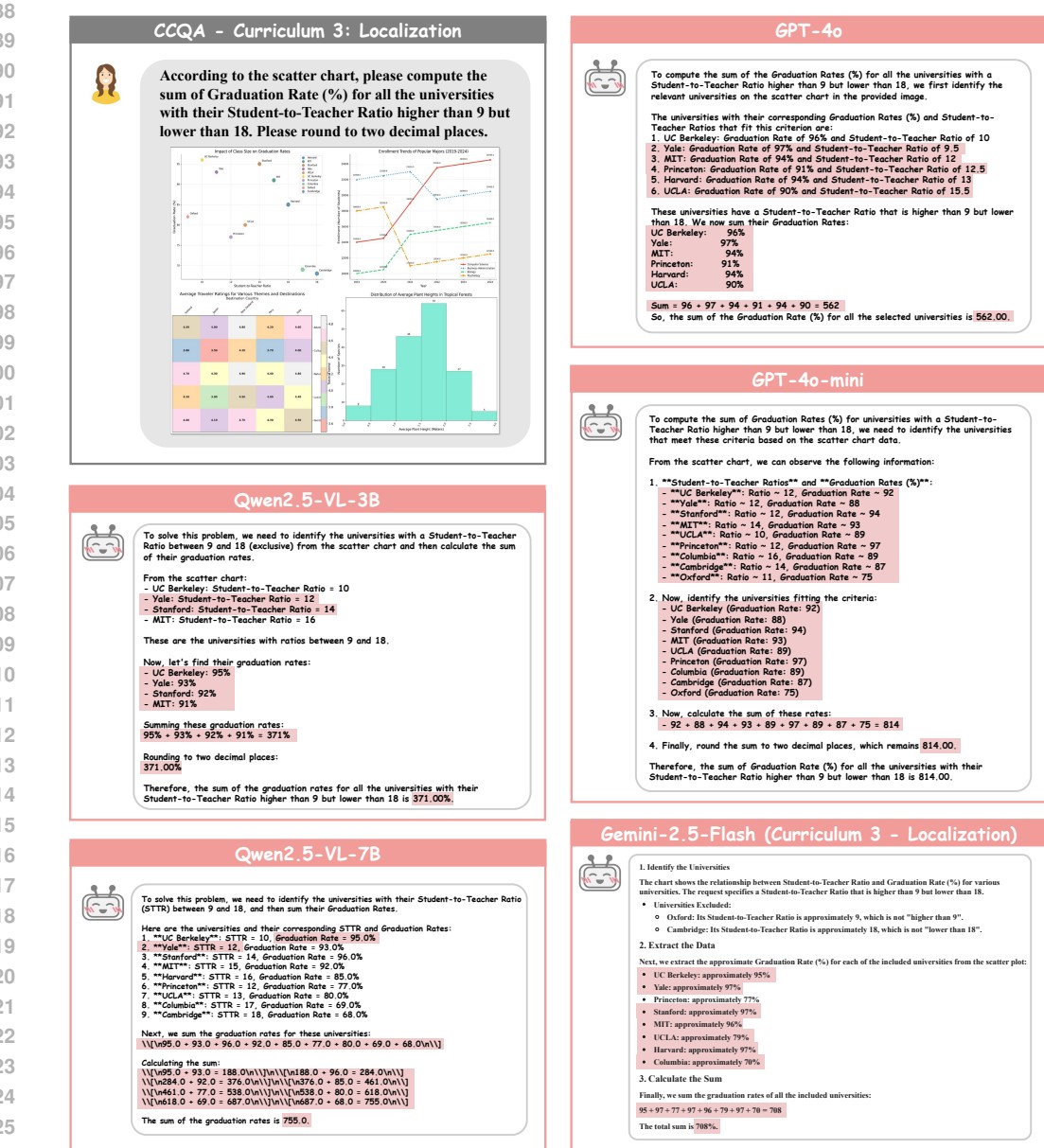

Figure 16: **Examples of Validation On Curriculum Level 3** ($D \geq 3$)**.** We validate our CCQA on *Tier 4* (§ B.3), i.e., *Curriculum Level 3 ($D \geq 3$)*, by prompting the model to generate CoT reasoning and final answer. CoT reasoning prompt is omitted in the input for clarify.

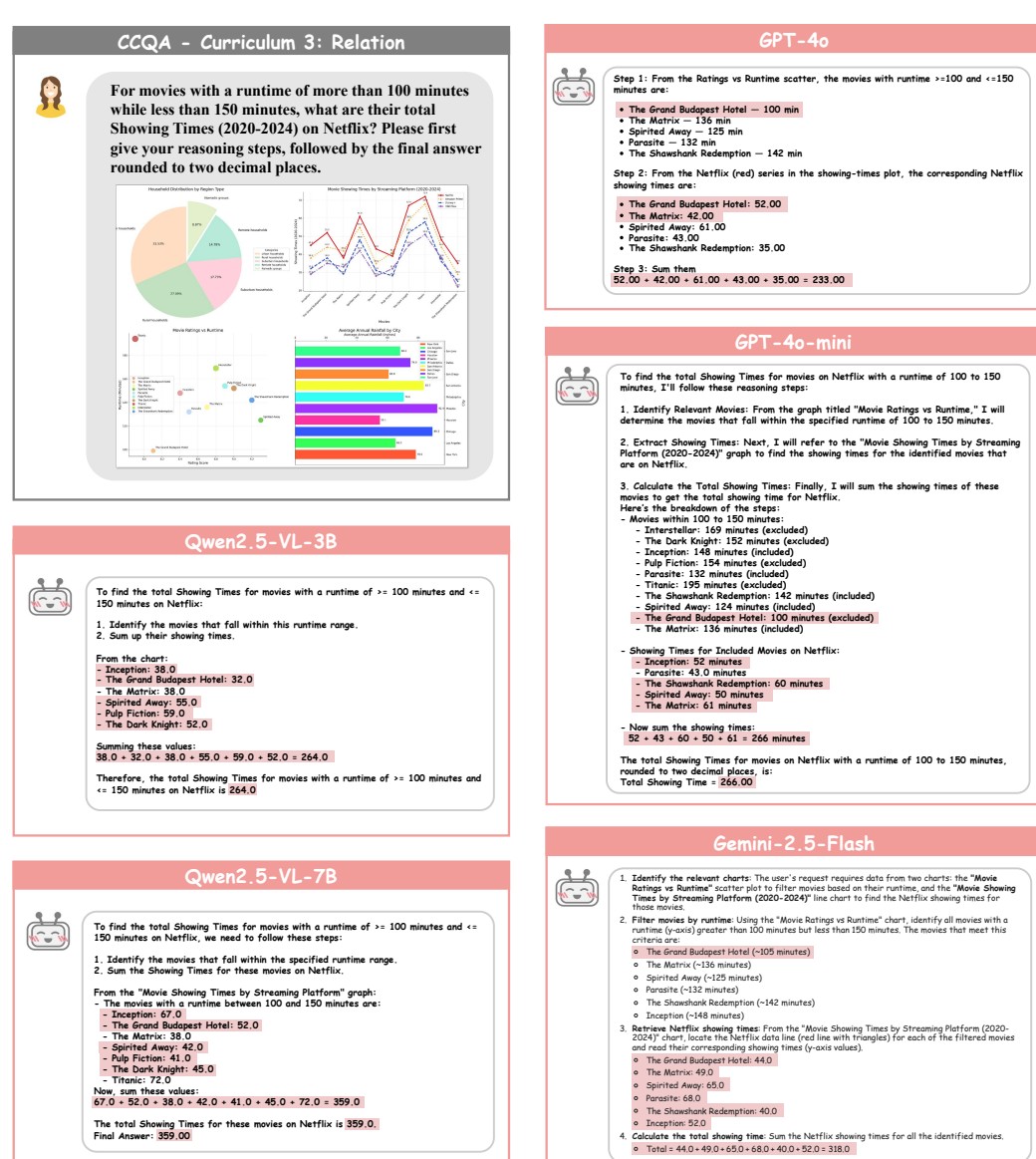

Figure 17: **Examples of Validation On Curriculum Level 3** ($D \geq 3$). We validate our CCQA on *Tier 5* (§ B.3), i.e., *Curriculum Level 3* ($D \geq 3$), by prompting the model to generate CoT reasoning and final answer. CoT reasoning prompt is omitted in the input for clarify.

## B  DATASET CONSTRUCTION

**Chart Metadata.** Extending our introduction to CCQA (§4), we elaborate on the seven types of charts, the domain categories of the source plotting data, and fundamental operators that support multi-layer nested functions, as summarized in Tables 4-5.

**Data Augmentation.** To effectively support curriculum learning with meta-learning insights (§B.4), we design a comprehensive set of chart-specific data augmentation strategies implemented through chart rendering functions. These augmentations introduce controlled variability in both the structural layout and visual presentation of charts, thereby enhancing model generalization across diverse chart types. Specifically, we consider the following transformations: (1) chart rotation at different angles (e.g., 0°, 30°, 45°, 60°, 75°, 90°); (2) orientation adjustments between vertical and horizontal layouts; (3) axis placement variations, such as shifting the $x$- and/or y-axis among left, right, top,

and bottom; (4) color setting across chart elements; (5) legend positioning (top, bottom, center, left, or right) and visibility; (6) label positions and visibili(ty such as axis labels, tick labels, and numeric annotations); and other augmentation strategies tailored for specific types of charts (e.g., 'explode' settings for pie charts, marker styles for scatter plots, etc.) Collectively, these augmentation strategies form a systematic approach for generating richly diverse chart appearances using a small set of metadata, ensuring robustness and adaptability of models trained under meta-learning supported curriculum learning.

Table 4: **Metadata of Chart Plotting.** We employ seven commonly used chart types across 30 different domain categories to construct the meta images for CCQA.

| Category | Description |
|---|---|
| **Chart Types** |  Bar, Histogram, Scatter, Line, Heatmap, Pie, Radar |
| **Domain Categories** | 1. Media & Entertainment
2. Geography & Demography
3. Education & Academia
4. Business & Industry
5. Major & Course
6. Animal & Zoology
7. Plant & Botany
8. Biology & Chemistry
9. Food & Nutrition
10. Space & Astronomy
11. Sale & Merchandise
12. Market & Economy
13. Sports & Athletics
14. Computing & Technology
15. Health & Medicine
16. Energy & Environment
17. Travel & Expedition
18. Arts & Culture
19. Communication & Collaboration
20. Language & Linguistics
21. History & Archaeology
22. Weather & Climate
23. Transportation & Infrastructure
24. Psychology & Personality
25. Materials & Engineering
26. Philanthropy & Charity
27. Fashion & Apparel
28. Parenting & Child Development
29. Architecture & Urban Planning
30. Gaming & Recreation |

Table 5: **Foundational Operators.** Our CCQA incorporates 12 basic operators to query different aspects of chart components, facilitating comprehensive understanding of each chart elements.

| Operator | Description |
|---|---|
| **Read** | Read or estimate the value of chart component(s) that meet given requirement(s) |
| **Statistics - Sum/Mean/Median** | Calculate the sum/mean/median of a group of chart components that meet given requirement(s) |
| **Statistics - Count** | Count the number of chart components that meet given requirement(s) |
| **Extrema - Value - Min/Max** | Calculate the minimum/maximum value (which may be combined with nested functions, *e.g.,* the minimum mean value of two groups of chart components) of chart components that meet given requirement(s) |
| **Extrema - Position** | Localize chart component(s) that meet given requirement(s), *e.g.,* the leftmost bar in the bar chart |
| **Sort - Ascending/Descending** | Sort a group of chart components that meet given requirement(s) |
| **Compare - Value/Diff/Position** | Compare the value/difference/position of two groups of chart components based on the given requirement(s) |
| **Filter** | Filter chart component(s) based on the given requirement(s) |
| **Threshold** | Identify chart component(s) based on the given threshold condition(s) |
| **Subset** | Identify the subset of chart component(s) that satisfy the specified requirement(s) |
| **Localization** | Localize specific chart components and/or subplots |
| **Relation** | Understand relations between or among different chart components and/or subplots |

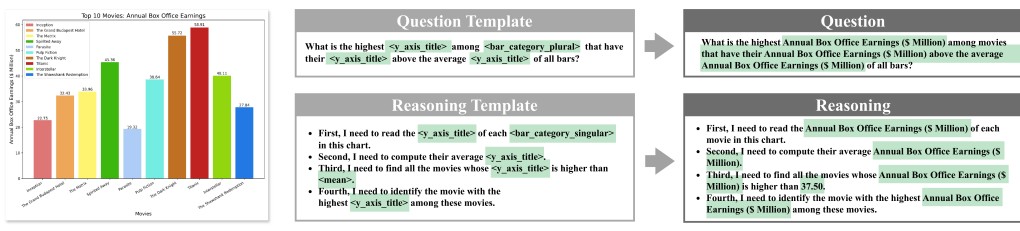

Figure 18: **From Template To CQA Data.** A template-based data generation example that illustrates how question and reasoning templates are converted to CQA data based on the chart data.

## B.1 DATA STRUCTURE

Our CCQA (§4) encompass seven basic chart types, including *bar chart, histogram, scatter plot, line chart, heatmap, pie chart, and radar chart* (Tab. 4). Only the chart plotting data (*i.e.*, the value and label of each chart component, along with the axis and image titles) are generated by GPT-4o (see an example in Fig. 19). We construct the CQA data of each chart type and curriculum level across 30 domain categories (Tab. 4). The number of samples for each chart type is influenced by chart features (e.g., scatter plots depend on both X- and Y-axis features, whereas heatmaps depend on cell values and labels), CQA types (Tab. 4, 5), and properties of the source plotting data (e.g., the number of bars, scatter points, or cells; variations in label angles; etc.). To ensure high data quality, we implement template-based CQA generation that guarantees not only the diversity of CQA tasks but also the accuracy and reliability of intermediate reasoning, visual grounding, and final answers.

Specifically, all question-answer pairs in CCQA, together with their corresponding reasoning steps and dynamic visual grounding coordinates, are generated using human-defined templates and functions (§4.1). An example is shown in Fig. 18 to illustrate the template-based CQA data generation process.

**GPT-4o Prompt For Plotting Data Generation**

Can you help with heatmap chart plotting data generation for my heatmap plotting?

**Data Example**
This plotting data example below is an input data dictionary for the first domain category "1 - Media & Entertainment".
```python
HEATMAP_DICT = {
    "heatmap_data": [[80.5, 20.5, 55.5, 90.0, 60.0], [30.5, 75.0, 45.5, 60.0, 80.0], [90.0, 10.5, 80.0, 20.0, 70.5], [80.0, 60.5, 30.0, 90.0, 59.5], [77.5, 80.5, 66.5, 25.5, 95.5]],
    "heatmap_category": {"singular": "rating score", "plural": "rating scores"},
    "x_labels": ["Titanic", "Star Wars", "The Lord of the Rings", "Snow White and the Seven Dwarfs", "The Lion King"],
    "y_labels": ["2005", "2010", "2015", "2020", "2025"],
    "x_label": "Movies",
    "y_label": "Rating Year",
    "image_title": "Correlation Between Movie Ratings and Rating Year",
}
```

**Your Task**
For the second domain category "2 - Geography & Demography", can you help generate a new plotting data using the same dictionary structure and data format?

**Important Requirements**
* Please generate the new plotting data specifically for the new domain category.
* Each data entry should be unique and should simulate real-world circumstances of the new domain category.
* Please be careful about `heatmap_category`, which will be used as how to generally call the heatmap cells. For example, `sentence=f"these heatmap cells represent different {HEATMAP_DICT['heatmap_category']['plural']}"` will become "these heatmap cells represent different *movie rating scores*".
* Please response in the following format:
```python
HEATMAP_DICT = <your new data dictionary>
```

```
HEATMAP_DICT = {
    "heatmap_data": [
        [70.5, 82.0, 65.0, 90.0, 75.0],
        [60.0, 55.0, 85.0, 80.0, 70.0],
        [55.0, 78.0, 88.0, 60.0, 65.0],
        [80.0, 68.0, 72.0, 85.0, 90.0],
        [95.0, 75.0, 60.0, 70.0, 80.0]
    ],
    "heatmap_category": {"singular": "urbanization index", "plural": "urbanization indices"},
    "x_labels": ["Tokyo", "New York", "Paris", "São Paulo", "Lagos"],
    "y_labels": ["1990", "2000", "2010", "2020", "2030"],
    "x_label": "Cities",
    "y_label": "Year",
    "image_title": "Urbanization Indices Over Time in Major Global Cities"
}
```

Figure 19: **Example of Plotting Data Generation Prompt.** We prompt GPT-4o to generate the source plotting data, which will be used as the input for chart drawing functions. This example is for the heatmap plotting data generation of the second domain category (Tab. 4).

## B.2 MULTI-LEVEL CURRICULUM

Implementing curriculum learning to progressively increase reasoning difficulty across three distinct levels (Fig. 5), each level targets at specific aspects of visual reasoning development (Tab. 5): *read*, *statistics*, *extrema*, *sorting*, *comparison*, *filtering*, *thresholding*, *subset constraints*, *localization*, and *relation*. The knowledge transfer between levels contributes to the increase of task complexity (§B).

**LEVEL 1: Foundational Single-Operation Reasoning.** LEVEL 1 establishes fundamental chart understanding capabilities (*reasoning depth*: $D_1 = 1$) with single-operation reasoning processes on single-plot charts:

$$\text{SINGLE}(x) = f(x) \tag{7}$$

where $f(\cdot)$ represents basic operations such as direct value reading, simple arithmetic, and elementary comparisons within a single-plot chart. Hereby, the ***reasoning depth*** of a CQA data sample, denoted as $D_l$, is defined as the number of nested operations for curriculum level $l$.

Accordingly, the template structure of LEVEL-1 CQA data follow the format:

$$\text{TEMPLATE}_1 = \{Q, \{R_t, B_t^*\}_{t=1}^{T_1}, A\} \tag{8}$$

where $Q$ is the question, $T_1 \geq 1$ is the number of reasoning steps, $R_t$ is the $t$-th reasoning step, $B_t^*$ is the corresponding ground-truth visual grounding with reasoning depth $D_1 = 1$, and $A$ is the final answer.

**LEVEL 2: Multi-Operation Reasoning.** LEVEL 2 introduces compositional reasoning (*reasoning depth*: $D_2 > 1$) through nested operations on single charts:

$$\texttt{SINGLE}(x) = F(f(x)) \tag{9}$$

where $F(\cdot)$ represents composite operations applied to $f(x)$, *e.g.*, $F(f(x)) = h(g(f(x)))$. This level requires models to perform sequential reasoning and visual grounding where each step builds upon previous computations. Consequently, LEVEL 2 templates extend to multi-step reasoning:

$$\text{TEMPLATE}_2 = \{Q, \{R_t, B_t^*\}_{t=1}^{T_2}, A\} \tag{10}$$

where $T_2 \geq 2$ is the number of reasoning steps, each reasoning step $R_t$ progresses through nested operations with reasoning depth $D_2 \geq 2$, and $B_t^*$ is the corresponding ground-truth visual grounding for the $t$-th step.

**LEVEL 3: Complex Multi-Chart Reasoning** LEVEL 3 represents the most challenging scenarios that involve complex reasoning ($D_3 \geq 2$) across multiple subplots and chart types:

$$\texttt{MULTI}(x) = \texttt{MULTI}(\texttt{SINGLE}_i(x)) \tag{11}$$

where $i$ conforms to $1 \leq i \leq n$ and $1 \leq n \leq$ *subplot_num*. LEVEL 3 templates thereby incorporate multi-step and cross-chart dependencies:

$$\text{TEMPLATE}_3 = \{Q, \{R_t, B_t^*, C_t\}_{t=1}^{T_3}, A\} \tag{12}$$

where $T_3 \geq 3$ is the number of reasoning steps, each reasoning step $R_t$ progresses through complex nested operations with reasoning depth $D_3 \geq 3$, $B_t^*$ is the corresponding ground-truth visual grounding, and $C_t$ indicates the chart index for the $t$-th reasoning step. Specifically, our multi-plot reasoning incorporates both localization ($n - 1$) and relation ($n > 1$) operations across multiple charts.

### B.3 FINE-GRAINED CURRICULUM TIERS

We construct our three-level curriculum dataset (Fig. 5) based on reasoning depth and chart complexity (§B.2). According to their fine-grained problem-solving difficulty, we categorize them into five curriculum tiers:

- **Tier 1: Curriculum Level 1** ($D = 1$). In *Tier 1*, all CQA data correspond to queries about single-plot chart image input. Reasoning is limited to one depth level, *i.e.*, single-function reasoning (Eq. 7, $D_1 \geq 1$), to derive the final answer.
- **Tier 2: Curriculum Level 2** ($D = 2$). In *Tier 2*, all CQA data correspond to queries about single-plot chart image input. Reasoning requires two depth levels, *i.e.*, constructed through two nested functions (Eq. 9, $D_2 \geq 2$), to derive the final answer.
- **Tier 3: Curriculum Level 2** ($D \geq 3$). In *Tier 3*, all CQA data correspond to queries about single-plot chart image input, with reasoning that involves three or more depth levels, *i.e.*, constructed through three or more nested functions (Eq. 9, $D_2 \geq 3$), to derive the final answer.
- **Tier 4: Curriculum Level 3 - Localization** ($D \geq 3$). In *Tier 4*, all CQA data correspond to queries about multi-plot chart image input, with reasoning that involves three or more depth levels, *i.e.*, constructed through three or more nested functions (Eq. 11, $D_3 \geq 3$), to derive the final answer. While different from single-plot charts, multi-plot CQA tasks in *Tier 4* involves the precise localization of target subplot(s) that directly yield the answer.

- **Tier 5: Curriculum Level 3 - Relation** ($D \geq 3$). CQA tasks in *Tier 5* are similar to the constitution of *Tier 4*, corresponding to queries about multi-plot charts with reasoning that involves three or more depth levels, *i.e.*, constructed through three or more nested functions (Eq. 11, $D_3 \geq 3$). The key distinction is that, while *Tier 4* emphasizes precise localization of target subplot(s), *Tier 5* additionally demands the modeling of relations across the identified subplots.

### B.4 META-LEARNING SUPPORTED CURRICULUM LEARNING

Our curriculum learning design (§3) is reinforced through meta-learning, which provides a principled way to structure both data and task complexity. Specifically, we leverage meta-learning through the following aspects:

1. **Domain diversity as meta-tasks.** We construct CCQA using 30 domain categories (Tab. 4), where each category contributes one source plotting data, and thus one chart image. This structured diversity provides a wide range of meta-tasks that expose MLLMs to domain-generalizable visual reasoning.

2. **Chart-type variability as meta-structures.** We employ 7 fundamental chart types (Tab. 4) to visualize the 30 domain datasets. Multiplying 30 plotting datasets by 7 chart types yields 210 unique chart-structure metadata, based on which the entire dataset is systematically constructed. This ensures that each domain is represented across diverse chart structures, promoting cross-task adaptation.

3. **Operator set as meta-functions.** To support multi-layer nested reasoning, we define 12 fundamental operators (Tab. 5). These operators serve as compositional primitives for constructing multi-level CQA tasks. By progressively increasing the depth of nesting, we control CQA difficulty level, thereby enabling MLLMs to gradually acquire higher-order reasoning capabilities.

4. **Decomposition as learning scaffolds.** Following decomposition insights (§A.1), we disentangle each task into *visual decomposition* (low-level chart components) and *reasoning decomposition* (singular operations across nested functions). This scaffolding allows MLLMs to incrementally learn fine-grained visual perception and step-wise logical reasoning, supporting the high-level composition of accurate chart understanding.

5. **Meta-learning for transferability.** Beyond dataset construction, our design leverages meta-learning to encourage transferability across chart types, domains, and reasoning depths. By repeatedly exposing MLLMs to varied meta-tasks with systematically controlled complexity, we enable them to acquire generalizable strategies rather than overfitting to particular chart types or reasoning templates. Our meta-learning implementation strengthens the robustness of curriculum learning by aligning it with principles of adaptation and generalization.

Together, these design principles ensure that our curriculum learning is not only systematic but also meta-learnable, allowing MLLMs to progressively integrate visual and reasoning competencies across tasks of increasing complexity.

## C VISUAL GROUNDING STRATEGIES

We propose three visual grounding strategies — *applied*, *boxed*, and *cropped* (Fig. 4, Tab. 6) — to enable *dynamic* visual focus navigation throughout the evolution of multi-turn reasoning. All three strategies follow the same **RVA** process where the model generates reasoning steps accompanied by grounded bounding box coordinates, while the "*dynamic*" nature refers to how the visual focus adaptively changes as the train of thoughts progresses. Each strategy implements a distinct grounding mechanism for directing the model's visual attention to corresponding image regions of focus while maintaining coherent reasoning flow. On the other hand, these strategies also represent different trade-offs between visual clarity, computational efficiency, and reasoning precision (Tab. 6).

Table 6: **Comparative Analysis of Visual Grounding Strategies.** We evaluate each grounding strategy across seven key dimensions that are significant for effective visual reasoning. ✓ indicates the strategy possesses the advantage, while ✗ indicates limitation. *Boxed* grounding achieves the best overall balance, *applied* grounding provides clear interpretability and highlighting with moderate trade-offs, and *cropped* grounding maximizes precision at the cost of computational efficiency and understanding straightforwardness.

| Method | Low Computation | High Precision | Full Context | No Occlusion | Multi-Region | Easy Integration | Easy Comprehension |
|---|---|---|---|---|---|---|---|
| applied | ✓ | ✗ | ✓ | ✗ | ✓ | ✓ | ✓ |
| boxed | ✓ | ✗ | ✓ | ✓ | ✓ | ✓ | ✓ |
| cropped | ✗ | ✓ | ✓ | ✓ | ✓ | ✗ | ✗ |

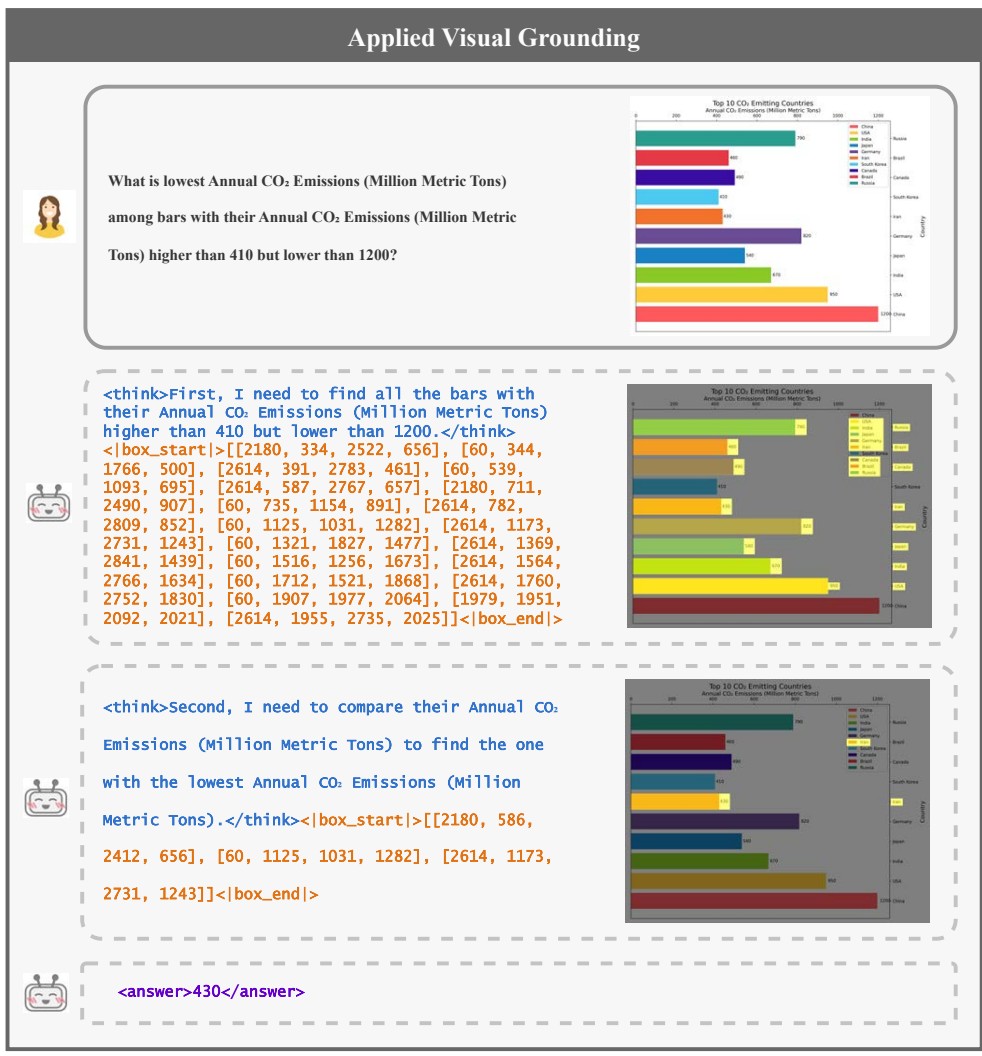

Figure 20: **Example of Applied Visual Grounding.** The *applied* visual grounding method directly accentuates the regions of focus through semi-transparent yellow highlighting overlays.

## C.1 APPLIED: GROUNDING THROUGH DYNAMIC VISUAL FOCUS HIGHLIGHTING

**Method.** The *applied* grounding strategy directly underlines the predicted regions of focus through semi-transparent yellow highlighting overlays. As the reasoning progresses, the yellow highlighting adaptively shifts to emphasize different visually focused regions mirroring each reasoning step, while preserving full visual context.

Specifically, our *applied* grounding strategy implements visual grounding through semi-transparent highlighting overlays that mask visual focuses:

$$I'_{vis,t} = I_{orig} \odot (1 - \kappa \cdot M_{focus,t}) + \kappa \cdot H_{yellow} \odot M_{focus,t} \tag{13}$$

where $I_{orig}$ is the original input image, $M_{focus,t}$ is the binary mask derived from the list of bounding boxes $\{B_{t,i}\}_{i=1}^{N_t}$ where each $B_{t,i} = [x_{min}, y_{min}, x_{max}, y_{max}]$ and $N_t$ is the number of bounding boxes at reasoning step $t$, $H_{yellow}$ is the highlight color (*i.e.,* yellow), $\kappa$ controls transparency, and $\odot$ denotes element-wise multiplication.

**Examples.** Fig. 20 shows an example for multi-turn CoT reasoning with *applied* visual grounding.

## C.2 BOXED: GROUNDING THROUGH DYNAMIC VISUAL BOX GUIDES

**Method.** The *boxed* grounding strategy straightforwardly guides visual attention by adding red rectangular borders to the focus regions. These red boxes dynamically relocate and resize along with the evolution of the reasoning chain, emphasizing current regions of focus through explicit visual boundaries.

Particularly, the border guides are generated by drawing rectangular outlines at the specified coordinates:

$$I'_{vis,t} = \text{BOX}(I_{orig}, \{B_{t,i}\}_{i=1}^{N_t}, C_{red}, \tau) \tag{14}$$

where $I_{orig}$ is the original input image, $\{B_{t,i}\}_{i=1}^{N_t}$ is the list of bounding boxes at reasoning step $t$ where each $B_{t,i} = [x_{min}, y_{min}, x_{max}, y_{max}]$, $\text{BOX}(\cdot)$ draws colored rectangular border lines for each specified regions of focus, $C_{red}$ is the border line color (*i.e.,* red), and $\tau$ is the thickness of border lines.

**Examples.** Fig. 21 shows an example for multi-turn CoT reasoning with *boxed* visual grounding.

## C.3 CROPPED: GROUNDING THROUGH DYNAMIC VISUAL FOCUS ZOOMING

**Method.** The *cropped* grounding strategy localizes corresponding regions of focus by zooming in, extracting and presenting the focused sub-regions as separate zoomed images alongside the full chart. As reasoning evolves, different cropped regions are dynamically generated and presented, enabling detailed examination of the specific components relevant to each reasoning step.

Therefore, the cropping operation extracts sub-regions using array indexing based on the bounding box coordinates:

$$I'_{vis,t} = \{I_{orig}, \{\text{CROP}(I_{orig}, B_{t,i})\}_{i=1}^{N_t}\} \tag{15}$$

$$\text{where } \text{CROP}(I_{orig}, B_{t,i}) = I_{orig}[y_{min} : y_{max}, x_{min} : x_{max}] \tag{16}$$

Here, $I_{orig}$ is the original chart image, $\{B_{t,i}\}_{i=1}^{N_t}$ is the list of bounding boxes at reasoning step $t$, and the model processes both the full context $I_{orig}$ and multiple zoomed crops $\{\text{CROP}(I_{orig}, B_{t,i})\}_{i=1}^{N_t}$ simultaneously.

**Examples.** Fig. 22 shows an example for multi-turn CoT reasoning with *cropped* visual grounding.

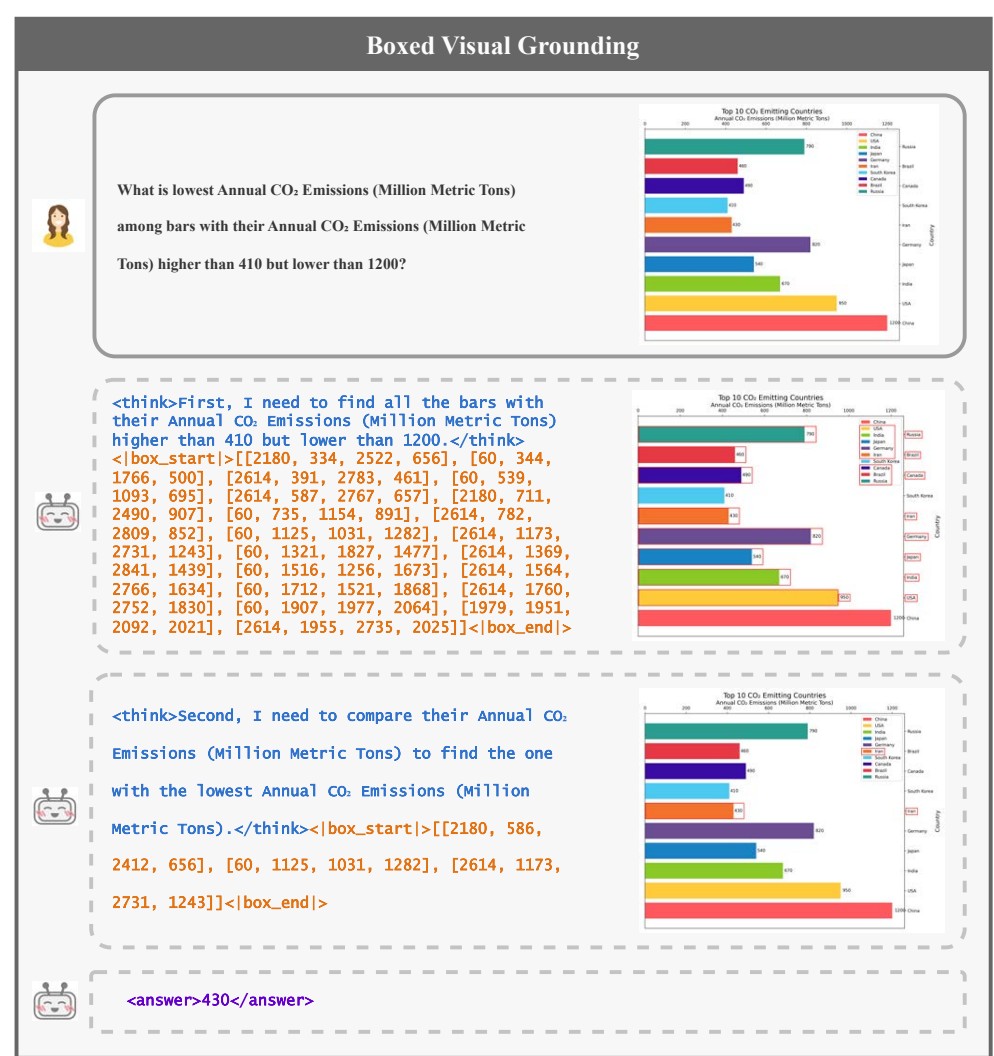

Figure 21: **Example of Boxed Visual Grounding.** The *boxed* visual grounding method directly accentuates the regions of focus through semi-transparent yellow highlighting overlays.

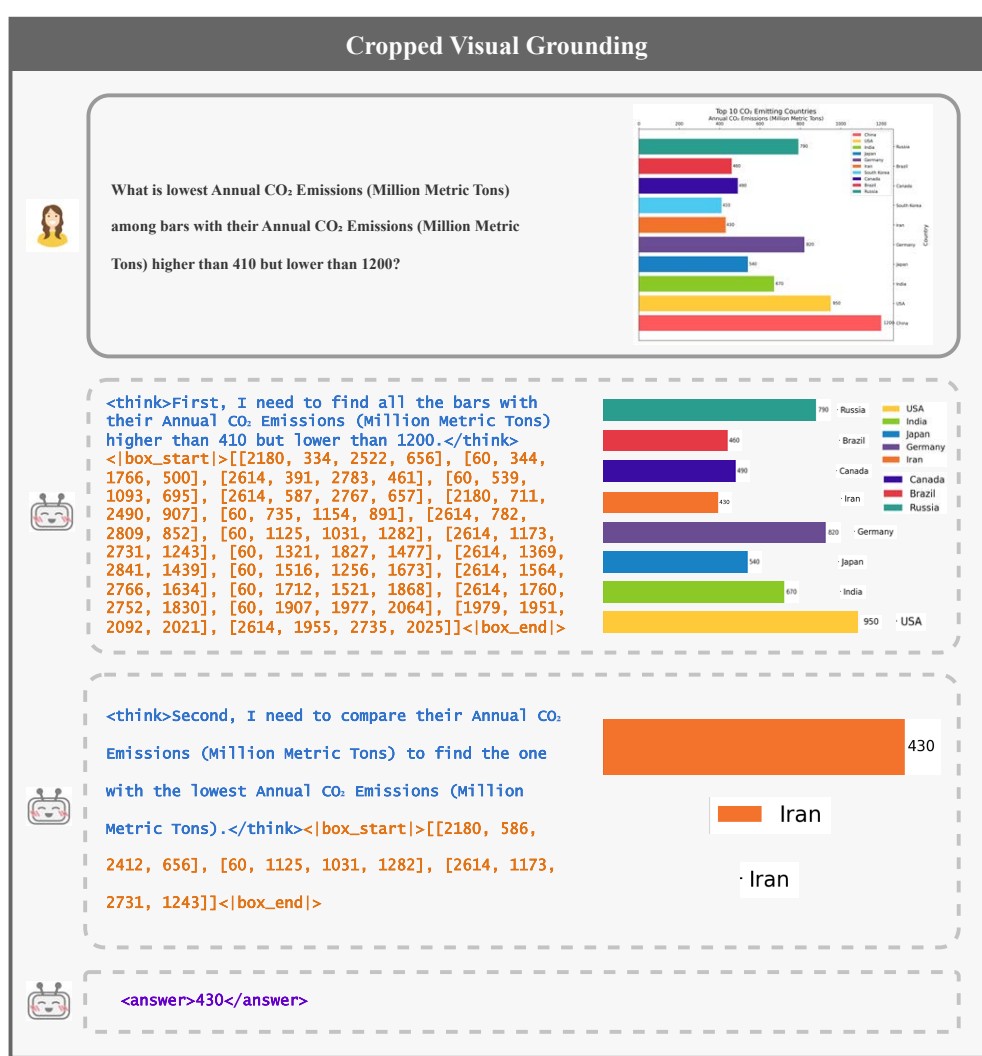

Figure 22: **Example of Cropped Visual Grounding.** The *cropped* visual grounding method directly accentuates the regions of focus through semi-transparent yellow highlighting overlays.

# D GENERATION MODE

## D.1 MODE A

Fig. 24 shows an example of generation mode **A**, where the model directly outputs the *answer* without intermediate reasoning and visual grounding.

## D.2 MODE VA

Fig. 23 shows an example of generation mode **VA**, through which the model first generates its intermediate *visual grounding* via *applied* grounding method, followed by its final *answer*. For clarity, the input instructions for **VA** generation are omitted in the figure. To save space, the user's intermediate responses are shown as smaller images on the right of each model response, corresponding to the model's response on the left in each turn of the multi-turn interaction.

### D.3 MODE RA

Fig. 25 shows an example of generation mode **RA**, through which the model first generates its intermediate *CoT reasoning*, followed by its final *answer*. The model is prompted to produce its CoT reasoning and final answer in a single-turn manner. For clarity, the input instructions of CoT reasoning and answering for **RA** mode are omitted in the figure.

### D.4 MODE RVA

Fig. 26 shows an example of generation mode **RVA**, where the model first produces its intermediate *reasoning* with *visual grounding*, followed by its final *answer*. Similar to Fig. 23, for clarity, the input instructions for **RVA** mode generation are omitted in the figure. To save space, the user's intermediate responses are shown as smaller images on the right of each model response, corresponding to the model's response on the left in each turn of the multi-turn **RVA** reasoning.

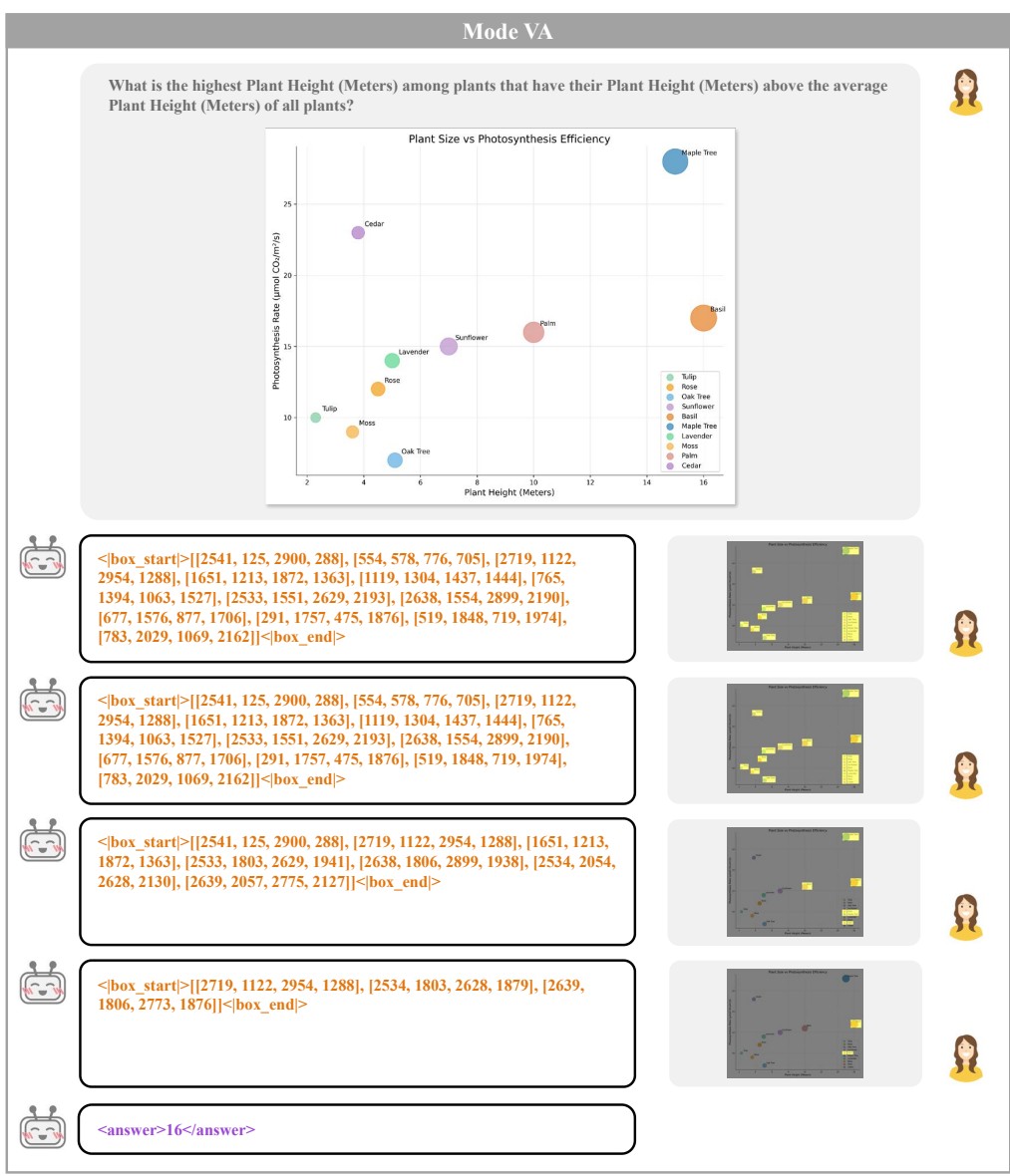

Figure 23: **Example of Generation Mode VA.** A CQA example resolved through generation mode **VA**.

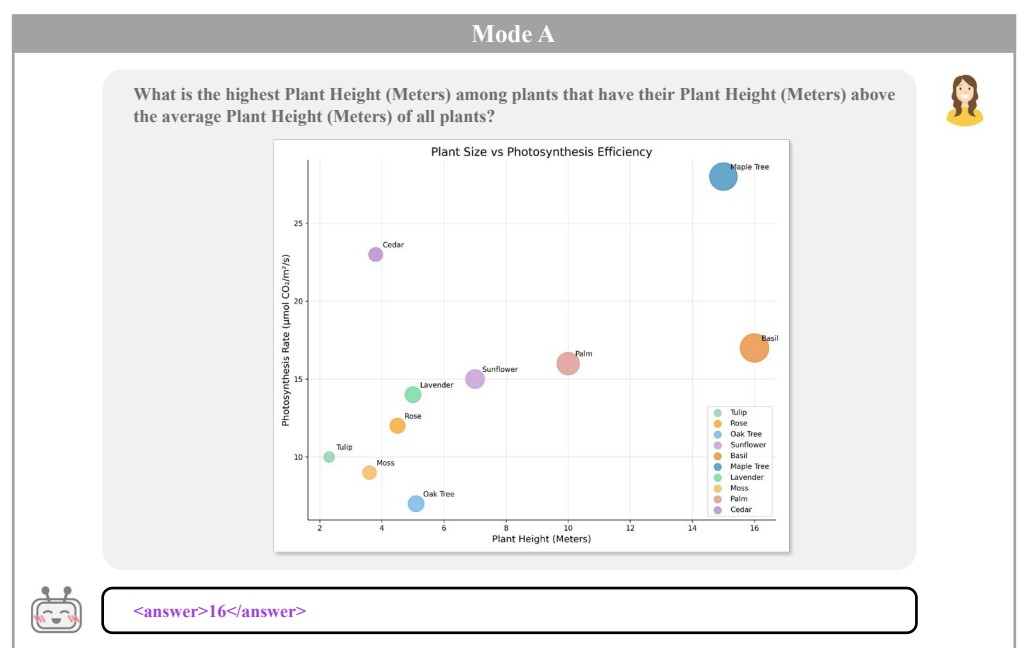

Figure 24: **Example of Generation Mode A.** A CQA example resolved through generation mode A.

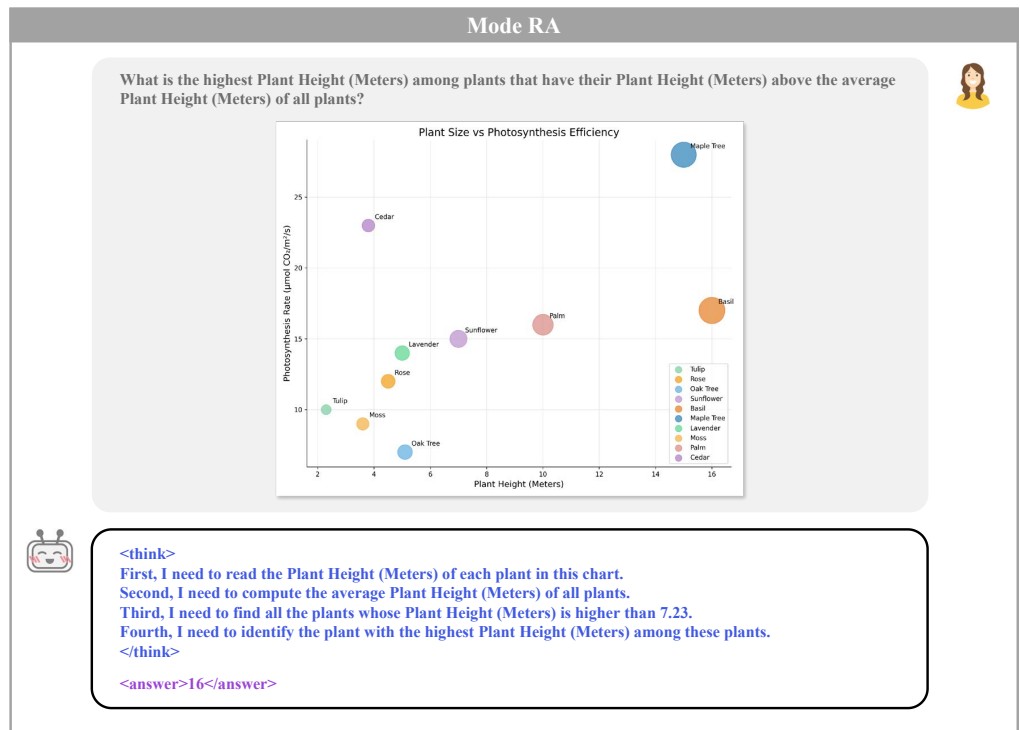

Figure 25: **Example of Generation Mode RA.** A CQA example resolved through generation mode RA.

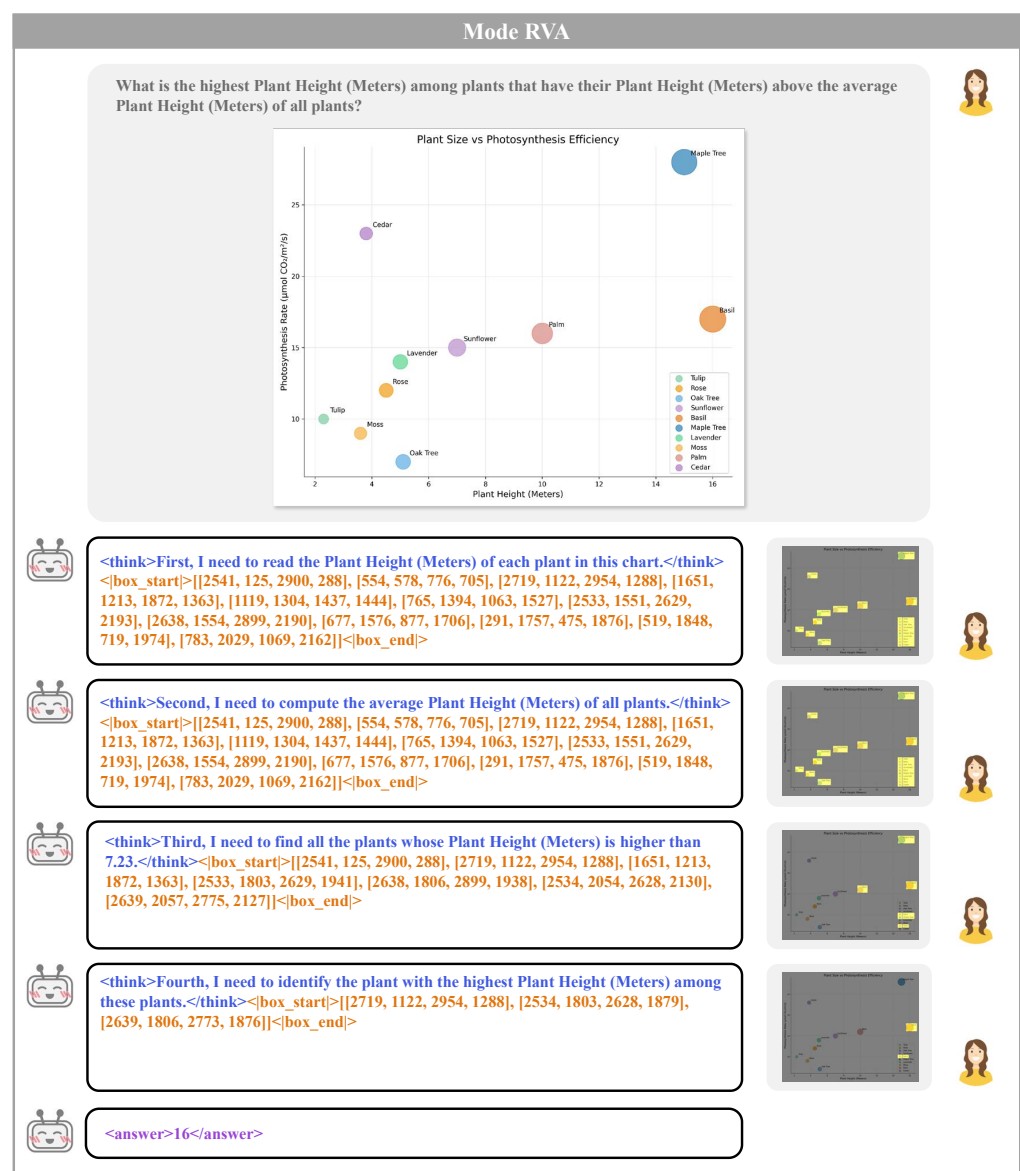

Figure 26: **Example of Generation Mode RVA.** A CQA example resolved through generation mode RVA.

# E EVALUATION METRICS

To elaborate more details in §5.2, our evaluation incorporates multiple complementary metrics to assess different aspects of model performance.

## E.1 EVALUATION OF ANSWERS

In pursuit of accurate evaluation on multimodal datasets that contain both multi-choice and free-form responses, we compute answer accuracy by comparing model outputs with their corresponding ground-truth answers. Aiming for more comprehensive assessment, we employ two complementary evaluation approaches: LLM-based judgment for semantic understanding and rule-based evaluation for systematic accuracy measurement. The overall accuracy score for each dataset is calculated as the mean accuracy across all test samples.

**LLM-Based Answer Evaluation.** For LLM-based answer evaluation, we employ GPT-4.1-mini as the *judge*, guided by the prompt shown in Fig. 27. Each model response undergoes LLM-as-judge evaluation to extract the model answer content, ensuring consistent comparison with ground truth. The *judge* performs a *True*-or-*False* assessment by evaluating whether the model response semantically matches the ground truth, accounting for variations in phrasing and presentation while maintaining semantic equivalence.

**Rule-Based Answer Evaluation.** To mitigate potential biases introduced by using LLMs as judges (§5.2), we complement the LLM-as-judge approach with a systematic rule-based evaluation (Algorithm 1). This rule-based method assesses answer accuracy through predefined parsing and judgment rules, incorporating both strict and relaxed error tolerance through four range criteria: *absolute accuracy* ($acc@0.0$) and three progressively relaxed thresholds ($acc@0.05$, $acc@0.1$, $acc@0.2$).

---

**Algorithm 1** Rule-Based Answer Evaluation with Tolerance Ranges

---

**Require:** Ground truth answer $gt$, predicted answer $pred$, choices $C$ (optional), tolerance ranges $R = \{0.0, 0.05, 0.1, 0.2\}$

**Ensure:** Accuracy scores $acc@r$ for each $r \in R$

1: $gt \leftarrow \text{CLEAN}(gt)$, $pred \leftarrow \text{CLEAN}(pred)$
2: $answer\_type \leftarrow \text{DETECTTYPE}(gt)$
3: **if** $gt = \emptyset$ **and** $pred = \emptyset$ **then**
4:     **return** $acc@r = 1.0$ for all $r \in R$
5: **end if**
6: **if** $answer\_type =$ ”multi-choice” **then**
7:     $gt\_list \leftarrow \text{PARSECHOICES}(gt, C)$
8:     $pred\_list \leftarrow \text{PARSECHOICES}(pred, C)$
9:     **if** $gt\_list = pred\_list$ **then**
10:         **return** $acc@r = 1.0$ for all $r \in R$
11:     **else**
12:         $match\_rate \leftarrow \frac{|gt\_list \cap pred\_list|}{|gt\_list|}$
13:         **for** $r \in R$ **do**
14:             $acc@r \leftarrow \mathbf{1}[match\_rate \geq r]$
15:         **end for**
16:     **end if**
17: **else if** $answer\_type \in \{$int, float$\}$ **then**
18:     $gt\_num \leftarrow \text{EXTRACTNUMBER}(gt)$
19:     $pred\_num \leftarrow \text{EXTRACT}(pred)$
20:     $acc@0.0 \leftarrow \text{EXTRACTNUMBER}(pred\_num, gt\_num)$
21:     **for** $r \in \{0.05, 0.1, 0.2\}$ **do**
22:         $lower \leftarrow gt\_num \times (1 - r)$
23:         $upper \leftarrow gt\_num \times (1 + r)$
24:         $acc@r \leftarrow \mathbf{1}[lower \leq pred\_num \leq upper]$
25:     **end for**
26: **else**
27:     $exact\_match \leftarrow \text{GRADEANSWER}(pred.lower(), gt.lower())$
28:     $substring\_match \leftarrow \mathbf{1}[|gt| > 5 \text{ and } gt.lower() \in pred.lower()]$
29:     $acc@r \leftarrow max(exact\_match, substring\_match)$ for all $r \in R$
30: **end if**
31: **return** $acc@r$ for all $r \in R$

---

## E.2 EVALUATION OF REASONING

To comprehensively evaluate model reasoning, we implement both micro- and macro-level assessments (§5.2). Our micro-level evaluation relies on five metrics (Eq. 17 - 21), providing the semantic similarity assessment of model reasoning. The final score ($acc@mic$) is the average across all five metrics. At the macro level ($acc@mac$), we leverage GPT-4.1-mini as the *judge*, which rates the quality of model reasoning on a $0 - 10$ scale based on three criteria: (1) *visual understanding and grounding*, (2) *logical coherence and multimodal integration*, and (3) *alignment with ground-truth reasoning*. While micro-level evaluation focuses on fine-grained similarity between ground-truth

and model reasoning, macro evaluation provides a holistic judgment of reasoning quality through LLM-as-judge (Fig. 28).

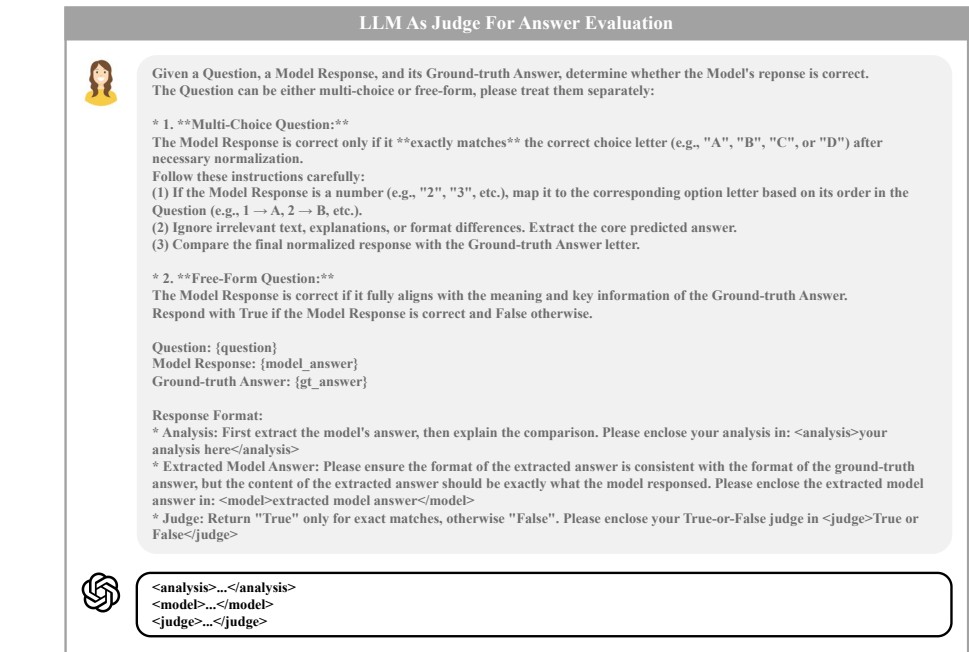

Figure 27: **LLM-As-Judge For Answer Evaluation.** We employ GPT-4.1-mini as the judge to assess model answer accuracy using the prompt shown in this figure.

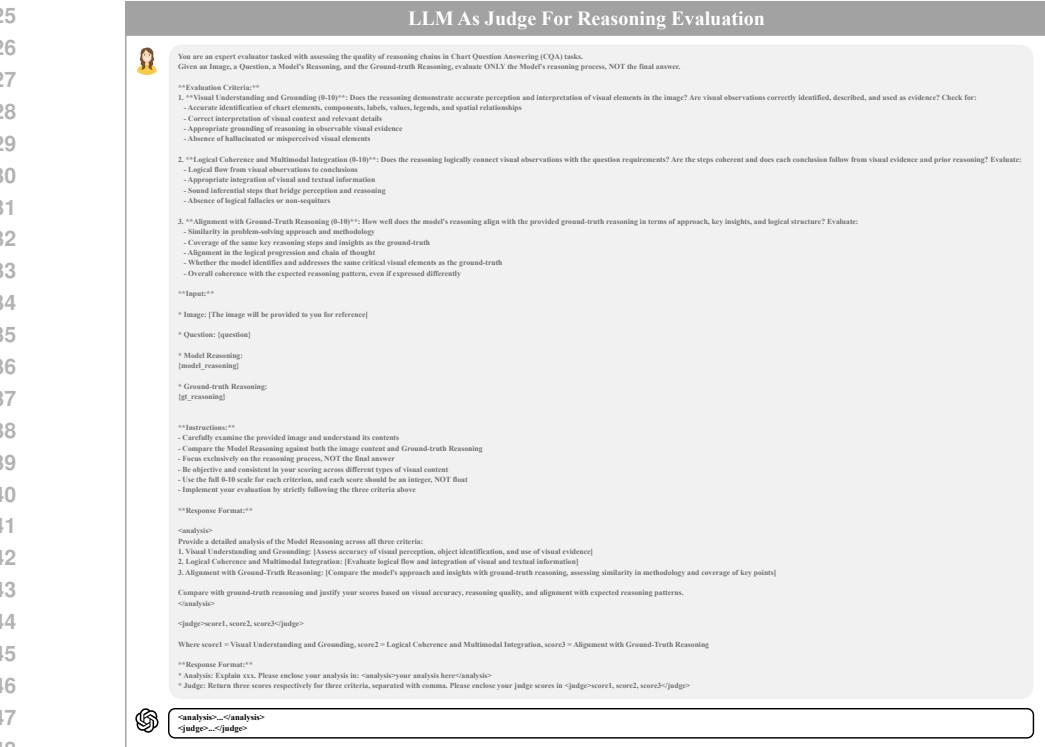

Figure 28: **LLM-As-Judge For Reasoning Evaluation.** We employ GPT-4.1-mini as the judge to evaluate model reasoning using the prompt shown in this figure. Prompt is restricted to smaller sizes to save space.

**Micro-Evaluation: Reasoning Similarity.**   We employ five metrics to measure the semantic similarity between ground-truth and model reasoning, including ROUGE-L (Eq. 17), BLEU (Eq. 18), METEOR (Eq. 20), BERTSCORE (Eq. 19), and COSINE SIMILARITY (Eq. 21).

$$\text{ROUGE} = \text{ROUGE-L} = \frac{2 \cdot P_{lcs} \cdot R_{lcs}}{P_{lcs} + R_{lcs}} \tag{17}$$

$$\text{BLEU} = \text{BLEU-4} = \text{BP} \cdot \exp\left(\sum_{n=1}^{4} w_n \log p_n\right) \tag{18}$$

where $P_{lcs}$ and $R_{lcs}$ are precision and recall of longest common subsequences.

$$\text{BERTSCORE} = \frac{1}{|\mathcal{S}_p|} \sum_{s_i \in \mathcal{S}_p} \max_{s_j \in \mathcal{S}_g} \frac{\mathbf{v}_i^T \mathbf{v}_j}{|\mathbf{v}_i||\mathbf{v}_j|} \tag{19}$$

where $\mathcal{S}_p$ and $\mathcal{S}_g$ are the sets of predicted and ground-truth reasoning tokens respectively, and $\mathbf{v}_i, \mathbf{v}_j$ are their corresponding BERT contextual embeddings.

$$\text{METEOR} = \frac{(1 + \eta_1) \cdot P_r \cdot R_r}{\eta_1 \cdot P_r + R_r} \cdot \left(1 - \eta_2 \cdot \left(\frac{c}{u_m}\right)^{\eta_3}\right) \tag{20}$$

$$\text{COSINE} = \frac{\mathbf{e}_p \cdot \mathbf{e}_g}{|\mathbf{e}_p||\mathbf{e}_g|} \tag{21}$$

where $P_r$ and $R_r$ are precision and recall of reasoning tokens, $u_m$ is the number of matched unigrams, $c$ is the number of chunks, and we define $\eta_1 = 0.9$, $\eta_2 = 0.5$, and $\eta_3 = 3$ as hyperparameters controlling the weight of recall, penalty magnitude, and penalty sharpness, respectively; and $\mathbf{e}_p$ and $\mathbf{e}_g$ are the embedding vectors of predicted and ground-truth reasoning steps respectively.

**Macro-Evaluation: Reasoning Quality.**   The quality of model reasoning is evaluated through LLM-as-judge assessment. Specifically, we employ GPT-4.1-mini as the *judge*, guided by the prompt shown in Fig. 28, to assign a quality score on a $0 - 10$ scale based on three criteria, including *visual understanding and grounding*, *logical coherence and multimodal integration*, and *alignment with ground-truth reasoning*. The final quality score for each dataset is calculated as the mean score across all test samples.

- **Criterion 1: Visual Understanding and Grounding.** Reasoning accuracy in identifying, interpreting, and grounding reasoning in visual elements, meanwhile without introducing hallucinated details
- **Criterion 2: Logical Coherence and Multimodal Integration.** Logical progression throughout the entire reasoning chain, with appropriate integration of multimodal information.
- **Criterion 3: Alignment with Ground-Truth Reasoning.** Consistency with ground-truth reasoning, especially in terms of problem-solving approach, key insights, and logical structure, even if expressed differently.

### E.3    EVALUATION OF VISUAL GROUNDING

As introduced in (§5.2), we employ two IoU variants, CIOU (Eq. 22) and GIOU (Eq. 23), as the primary evaluation metrics for visual grounding assessment.

$$\text{CIoU} = \text{IoU} - \frac{\rho^2(c_p, c_g)}{d^2} \tag{22}$$

$$\text{GIoU} = \text{IoU} - \frac{|A_c - A_u|}{A_c} \tag{23}$$

where $\rho^2(c_p, c_g)$ is the squared distance between predicted and ground-truth centroids, $d$ is the diagonal of the enclosing box, $A_c$ is the enclosing area, and $A_u$ is the union area.

### E.4 TRAINING METRICS

During training, we integrate *focal binary cross-entropy loss* (Eq. 24) with *Dice loss* (Eq. 26) to compute the mask loss (Eq. 6):

**Binary Cross-Entropy (BCE) Loss.** BCE loss ensures pixel-level grounding accuracy, and we implements *focal* BCE loss (Eq. 24) to emphasize multi-object grounding and address class imbalance:

$$\mathcal{L}_{\text{BCE}} = -\frac{1}{HW} \sum_{h=1}^{H} \sum_{w=1}^{W} w_{h,w}[M_{h,w}^* \log(M_{h,w}) + (1 - M_{h,w}^*) \log(1 - M_{h,w})] \qquad (24)$$

where $M^*$ is the ground-truth binary mask, $M$ is the predicted mask derived from parsed bounding boxes, and $w_{h,w}$ is the focal weight (Eq. 25) defined as:

$$w_{h,w} = \begin{cases} \lambda(1 - p_{h,w})^\phi & \text{if } M_{h,w}^* = 1 \\ (1 - \lambda)p_{h,w}^\phi & \text{if } M_{h,w}^* = 0 \end{cases} \qquad (25)$$

where $p_{h,w} = M_{h,w}^* \cdot M_{h,w} + (1 - M_{h,w}^*) \cdot (1 - M_{h,w})$ is the probability of the correct class, $\lambda$ is the class weighting factor, and $\phi$ is the focusing parameter.

**Dice Loss.** We leverage *Tversky* loss (Salehi et al., 2017), a generalized form of Dice loss designed to address class imbalance in visual grounding by asymmetrically weighting false positives and false negatives:

$$\mathcal{L}_{\text{Dice}} = \frac{\sum_{h,w} M_{h,w} M_{h,w}^* + \epsilon}{\sum_{h,w} M_{h,w} M_{h,w}^* + \delta \cdot \sum_{h,w}(1 - M_{h,w})M_{h,w}^* + (1 - \delta) \cdot \sum_{h,w} M_{h,w}(1 - M_{h,w}^*) + \epsilon} \qquad (26)$$

where $\delta$ is the asymmetric weighting parameter that controls the balance between precision and recall, and $\epsilon$ is a smoothing factor to prevent division by zero. When $\delta = 0.5$, $\mathcal{L}_{\text{Dice}}$ reduces to standard Dice loss, while values of $\delta < 0.5$ emphasize recall and values of $\delta > 0.5$ emphasize precision.

For reasoning, we compute the SIMILARITY (Eq. 27) as a weighted combination of Eq. 17 and Eq. 18:

$$\text{SIMILARITY} = \mu_2 \cdot \text{ROUGE} + (1 - \mu_2) \cdot \text{BLEU} \qquad (27)$$

**Visual Grounding Metrics.** For visual grounding evaluation, we compute Intersection over Union (IoU) variants CIOU (Eq. 22) and GIOU (Eq. 23), as adapted from our evaluation metrics (§5.2). The final $m$IoU (Eq. 28) is therefore calculated as a weighted combination of Eq. 22 and Eq. 23:

$$m\text{IoU} = \mu_3 \cdot \text{CIOU} + (1 - \mu_3) \cdot \text{GIOU} \qquad (28)$$

where $\mu_3$ is the strength parameter to balance between CIOU and GIOU.

**Combined Performance Metric.** We define a unified metric that balances all aspects:

$$\text{COMBINED} = w_1 \cdot \text{ACCURACY} + w_2 \cdot \text{SIMILARITY} + w_3 \cdot m\text{IoU} \qquad (29)$$

where $w_1$, $w_2$, and $w_3$ are balancing weights for accuracy, similarity, and grounding performance respectively.

### E.5 EVALUATION MODE

Furnishing models with the capabilities to reason through dynamic visual grounding, we employ different generation modes (§5.1) to support comparable evaluations:

- **Mode A**: *Answer-Only* mode where MLLMs are prompted to directly generate the final answer.
- **Mode RA:** *Reason-Answer* mode where MLLMs first go through the intermediate reasoning process, followed by the final answer.

- **Mode VA:** *Vision-Answer* mode where MLLMs first generate their visual grounding coordinates, followed by the final answer.
- **Mode RVA:** *Reason-Vision-Answer* mode where MLLMs first go through the reasoning process with dynamic visual grounding, and then generate the final answer.

## F IN-DEPTH ANALYSIS

### F.1 IMPLEMENTATION DETAILS.

We train each model for 3 epochs with an initial learning rate $lr = 1e - 4$ using `cosine` scheduler. The ratio of training and validation is set to `train:val=9:1`. Employing NVIDIA 80G H100 GPUs, our model training is powered by LoRA for memory efficiency. For hyperparameter settings, we define mask loss weight $\alpha = 0.5$, BCE weight $\beta = 0.8$, Dice weight $\gamma = 0.2$, and combined metric weights $w_1 = 0.4$, $w_2 = 0.3$, $w_3 = 0.3$, respectively. Implementing different generation modes, *reasoning* (**R**) is enclosed within `<think></think>`, *visual grounding coordinates* (**V**) are enclosed within `<|box_start|><|box_end|>`, and the *final answer* (**A**) is enclosed within `<answer></answer>`.

### F.2 GROUNDING METHOD & COMPUTATION COST

Employing zoom-in visual grounding, the *cropped* grounding method requires substantially larger memory at the same resolution. To mitigate this cost, we reduce the training resolution of *cropped* grounding to $128 \times 128$, thereby maintaining a comparable computational overhead. Despite the resolution degradation, reasoning with **cropped** visual grounding achieves notably higher accuracy than the baseline (up to $4.72\%$ improvement on CCQA) and performs competitively with the other two grounding methods (Tab. 2). These results highlight the effectiveness of zoom-in visual enhancement, albeit at the expense of increased computational cost when aiming for higher performance.

Table 7: **The Computation & Configuration Of Different Grounding Method.** This table summarizes the visual computation requirements and parameter configuration.

| Grounding Method | Resolution | $D_{max}$ | $T_{max}$ | $\alpha$ | $\beta$ | $\gamma$ |
|:---:|:---:|:---:|:---:|:---:|:---:|:---:|
| **Applied** | $448 \times 448$ | 4 | 5 | 0.5 | 0.8 | 0.2 |
| **Boxed** | $448 \times 448$ | 4 | 5 | 0.5 | 0.8 | 0.2 |
| **Cropped** | $128 \times 128$ | 4 | 5 | 0.5 | 0.8 | 0.2 |

### F.3 THE ROLE OF VISUAL GROUNDING: FROM EXTRINSIC ASSISTANCE TO INTRINSIC ABILITIES

A critical finding from our experiments reveals the fundamental distinction between the utility of multi-turn visual reasoning during training versus inference (Tab. 8). While incorporating visual grounding in the training process significantly enhances models' intrinsic visual reasoning capabilities, directly applying the same multi-turn approach during inference can paradoxically degrade performance due to error accumulation (Fig. G.2 & §G.1).

**Training Benefits of Visual Grounding.** Our curriculum learning approach with visual grounding supervision effectively teaches models to develop stronger intrinsic representations for chart understanding. By learning to align reasoning steps with visual focuses during training, models internalize the ability to focus on relevant image components, leading to improved performance even when generating direct answers without explicit visual grounding steps.

**Inference Challenges with Multi-Turn Visual Grounding.** On the other hand, when models are required to explicitly generate visual grounding coordinates during inference (i.e., *Mode RVA*), performance degrades in comparison with direct answer generation (*Mode A*) and reasoning without grounding (*Mode RA*). This degradation stems from two primary factors: (1) *Cumulative grounding*

*errors*: Inaccurate bounding box predictions in early reasoning steps propagate and compound errors in subsequent steps; (2) *Reasoning-grounding misalignment*: Discrepancies between intended visual focus and actual predicted coordinates lead to reasoning based on incorrect visual regions.

**Power of Intrinsic Visual Reasoning Capabilities.** Results in Tab. 8 demonstrate that visual grounding serves as an effective *training signal* rather than an *inference mechanism*. Our curriculum learning with visual supervision enables models to learn better intrinsic visual-textual alignments, which manifest as improved performance in direct answer generation Tab. 2. However, explicitly requiring visual grounding during inference introduces additional complexity and error sources that outweigh the potential benefits. Nevertheless, compared with baselines, our finetuned models manage to achieve remarkably higher performance in not only *Mode* **A**, but also *Modes* **RA**, **VA**, and **RVA**.

# G    MULTI-TURN REASONING WITH DYNAMIC VISUAL GROUNDING

Table 8: **Performance Evaluation for RVA Mode Inference.** Employing the set of evaluation metrics (§ 5.2), we assess model reasonnig, visual grounding, and final answer, respectively.

| Model | Size | Level 1 | | | | Level 2 | | | | Level 3 | | | |
|---|---|---|---|---|---|---|---|---|---|---|---|---|---|
| | | Reasoning | | Grounding | Answer | Reasoning | | Grounding | Answer | Reasoning | | Grounding | Answer |
| | | acc@mac | acc@mic | mIoU | acc@mac | acc@mac | acc@mic | mIoU | acc@mac | acc@mac | acc@mic | mIoU | acc@mac |
| Baselines | | | | | | | | | | | | | |
| GPT-4o | - | 53.17 | 51.02 | 22.03 | 50.00 | 53.32 | 53.60 | 13.01 | 24.00 | 50.20 | 46.86 | 11.14 | 21.50 |
| Qwen2.5-VL | 3B | 44.90 | 39.07 | 37.12 | 36.00 | 38.72 | 40.08 | 29.73 | 14.00 | 32.17 | 35.47 | 27.43 | 12.00 |
| | 7B | 48.53 | 41.70 | 48.17 | 45.00 | 40.49 | 40.68 | 32.25 | 21.50 | 35.82 | 38.10 | 32.26 | 17.50 |
| Ours | | | | | | | | | | | | | |
| Applied | 3B | 50.50 | 53.63 | 47.17 | 50.00 | 45.79 | 45.66 | 40.59 | 19.00 | 41.95 | 38.76 | 34.60 | 15.00 |
| (Qwen2.5-VL) | 7B | 56.13 | 54.01 | 57.55 | 52.00 | 49.87 | 47.17 | 45.72 | 23.00 | 46.05 | 38.83 | 39.90 | 19.50 |
| Boxed | 3B | 50.23 | 49.90 | 45.55 | 46.00 | 40.70 | 42.74 | 38.11 | 14.50 | 36.98 | 36.97 | 33.37 | 13.50 |
| (Qwen2.5-VL) | 7B | 51.27 | 48.20 | 50.88 | 49.00 | 42.97 | 43.44 | 43.09 | 22.50 | 41.33 | 37.45 | 34.52 | 18.00 |
| Cropped | 3B | 42.17 | 46.68 | 51.22 | 39.00 | 40.52 | 42.69 | 45.82 | 14.50 | 39.63 | 35.86 | 34.27 | 13.00 |
| (Qwen2.5-VL) | 7B | 50.33 | 50.79 | 51.33 | 47.00 | 44.97 | 43.75 | 50.10 | 24.00 | 42.43 | 38.51 | 39.19 | 19.00 |

## G.1    CHALLENGE: INFERENCE WITH VISUALLY GROUNDED REASONING

We leverage CCQA, randomly selecting 500 samples to evaluate model inference through **RVA** mode (§5.1). Training MLLMs with explicit reasoning and visual grounding as intermediate outputs effectively enhances model's intrinsic visual reasoning capabilities (§6.1). This step-by-step visual reasoning guides the model to decompose complex tasks into structured reasoning chains through dynamic attention grounding. With intermediate grounding naturally supporting more coherent reasoning trajectories, this in turn enhances model's ability to establish interleaved thinking-perception correspondences. Aligning with human visual reasoning, *decomposed* reasoning chains effectively help models to develop and strengthen their intrinsic visual reasoning capabilities.

Different from learning, during inference, human visual reasoning is rather a *composed* process that interleaves logical reasoning with visual comprehension, while *compositing* all intermediate steps into a coherent chain of thought. In contrast, inference in **RVA** exposes the fragility of step-wise generation: once an intermediate step is flawed, whether by incorrect calculation or inaccurate visual comprehension, the error propagates through the chain, breaking the balance between perception and reasoning that eventually leads to incorrect final answers (§G.1). Therefore, *it can be an effective way of learning, while may not be as useful in inference*.

Tab. 8 summarizes the evaluation results of **RVA** inference. Fintuned models achieve noticeable improvements across reasoning (up to 10.23% absolute gain), grounding (up to 9.38% absolute gain), and answering (up to 14% absolute gain). Beyond these numerical results, qualitative inspection (§G.3) reveals distinct behavioral patterns where training with step-by-step visual grounding encourages systematic reasoning chains with sharper object localization, showcasing stronger alignment with human-like reasoning trajectories.

Despite these improvements, however, the answering performance remains lower than that of ❀ CURV when using the same base model and grounding method (Tab. 2). This indicates that, while **RVA** training can effectively enhance intrinsic visual reasoning capabilities, **RVA** inference magnifies the vulnerability to intermediate error accumulation.

Build upon our discussions above, we present the challenge for MLLMs in CQA through mode **RVA**. Although **RVA** training notably enhances model's intrinsic visual reasoning capabilities,

RVA inference reveals critical limitations. During RVA inference, cumulative intermediate errors can cascade through the reasoning chain, ultimately compromising the model's ability to produce correct answers. For example, Fig. 29 shows a CQA example that is responded to by GPT-4o and Qwen2.5-VL-7B, respectively. Both GPT-4o and Qwen2.5-VL-7B fail to generate the correct answer due to their disrupted reasoning chains that evolve through their incorrect visual grounding.

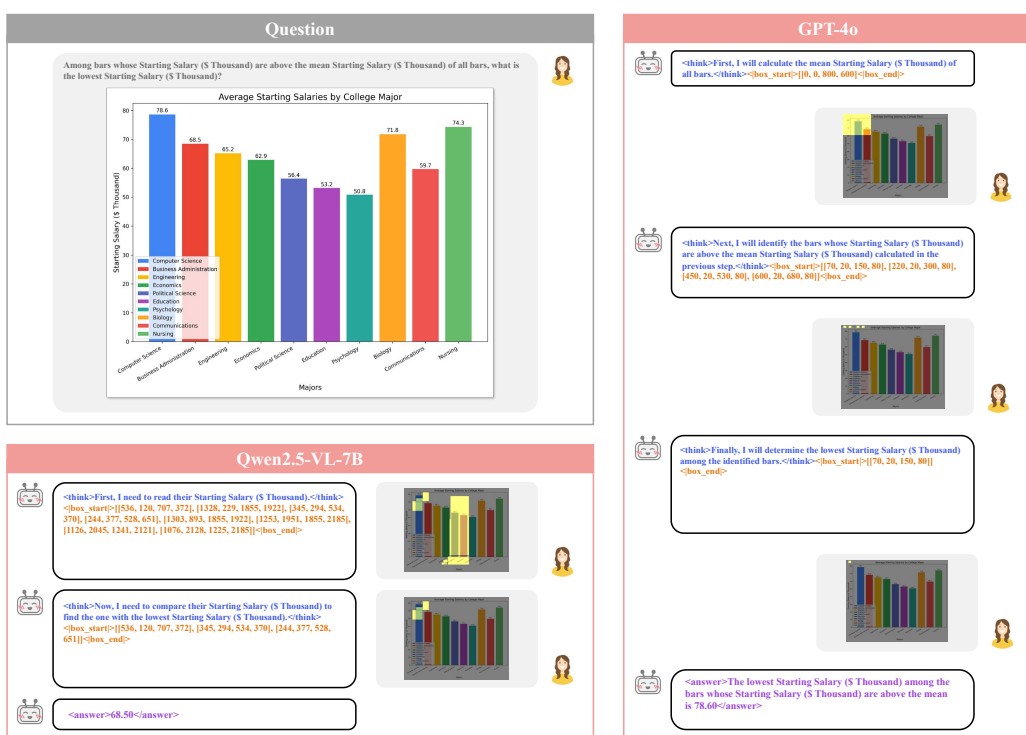

Figure 29: **Challenge of Inference in Mode RVA.** Tested on GPT-4o and Qwen2.5-VL, this example illustrates the challenge MLLMs face in performing inference in RVA.

## G.2 INFERENCE FAILURE

Fig. 32 illustrates examples of model inference failures in RVA. In both cases, the model fails to properly ground its reasoning in the chart, leading to inaccurate extraction and misinterpretation of visual information. Arising in early reasoning steps, these visual comprehension inaccuracies can propagate through the reasoning chain, ultimately resulting in incorrect question answering.

## G.3 INFERENCE SUCCESS

**Example 1 - Mode A:**  Figures 30 & 31 exhibit examples on chart question answering in mode A, where the baseline Qwen2.5-VL-7B fails to generate the correct answer, while ❀ CURV (Qwen2.5-VL-7B) finetuned through *applied* grounding succeeds. Fig. 30 (a) is a simple value reading problem ($D = 1$), where the baseline model fails to localize the exact queried chart component. Fig. 30 (b) consists of two nested functions ($D = 2$), where the baseline model fails to localize the queried bar in the given subset of countries. Fig. 30 (c) further enhance the CQA complexity ($D = 3$), involving three nested functions across reasoning, visual grounding, and interleaved calculation that the baseline model fails to correctly response. Different from Fig. 30 that query about a single chart, each CQA sample in Fig. 31 involves multiple charts that significantly complicates question answering. The baseline model fails in Fig. 30 (a) ($D = 3$) as it requires the localization of the exact chart subplot, the required subset, as well as the Y-axis value reading. Fig. 30 (b) increases the CQA difficulty ($D = 4$) by including not only accurate localization of chart components, but also extremia comparison of both bar values and spatial positions. Fig. 30 (c) ($D = 5$) presents

further enhanced complexity by involving relations across different charts. This relational chart understanding making the problem solving more challenging, unveiling the significance of accurate visual reasoning in tackling complex CQA tasks.

**Example 2 - Mode RVA:** Fig. 33 presents two examples of successful inference in RVA mode. The bar chart example on the left shows reasoning with accurate visual grounding. The heatmap example on the right shows a case where the grounding is not exact but falls close to the regions of focus, also leading to the correct answer.

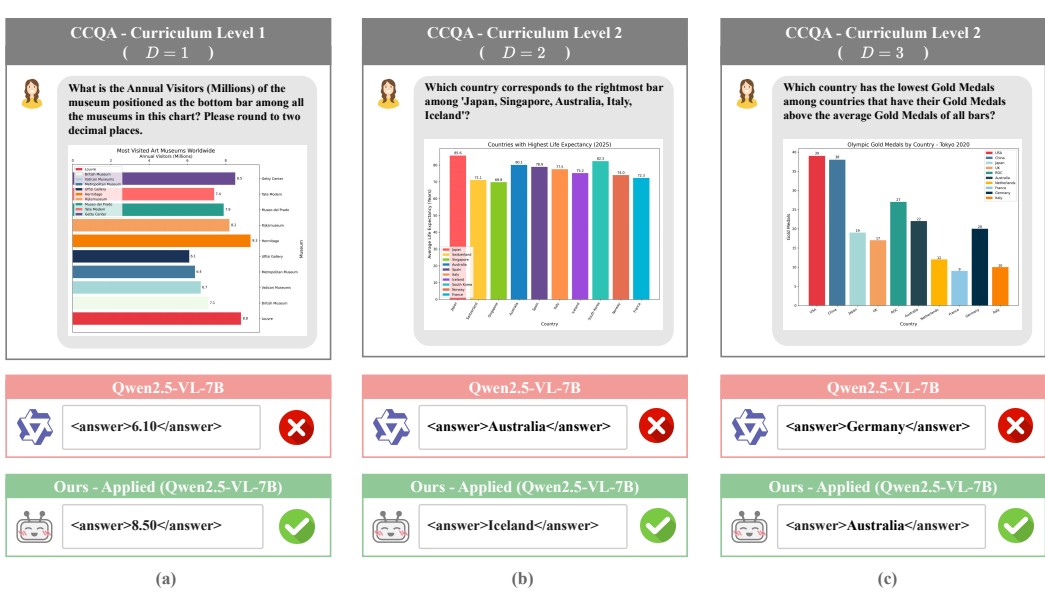

Figure 30: **Success Examples On Curriculum Levels 1-2.** This figure shows three examples on curriculum levels 1-2 of CCQA, where the baseline Qwen2.5-VL-7B fails while our ❀ CURV using *applied* grounding succeeds.

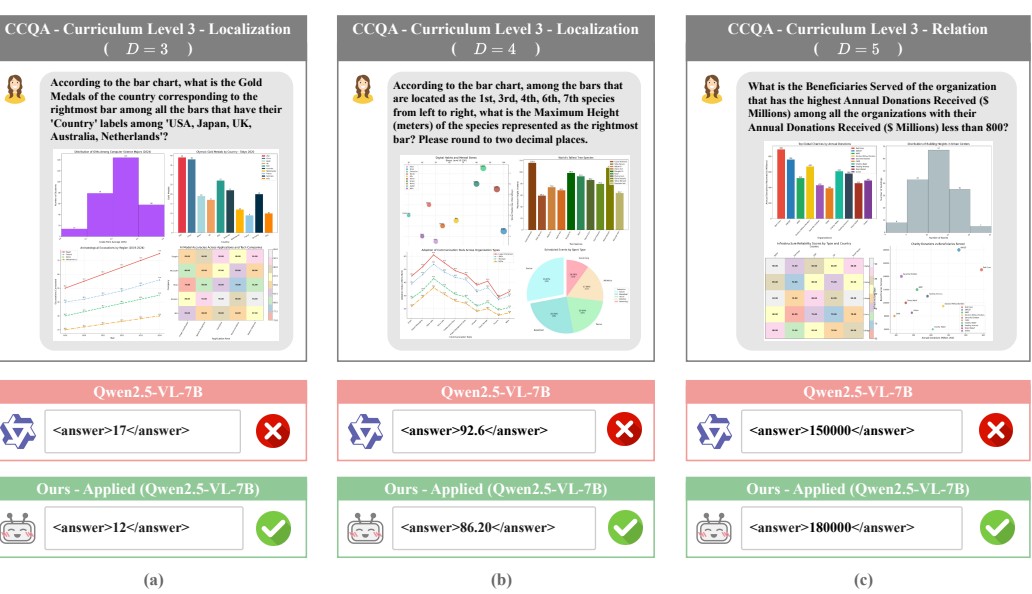

Figure 31: **Success Examples On Curriculum Level 3.** This figure shows three examples on the curriculum level 3 of CCQA, where the baseline Qwen2.5-VL-7B fails while our ❀ CURV using *applied* grounding succeeds.

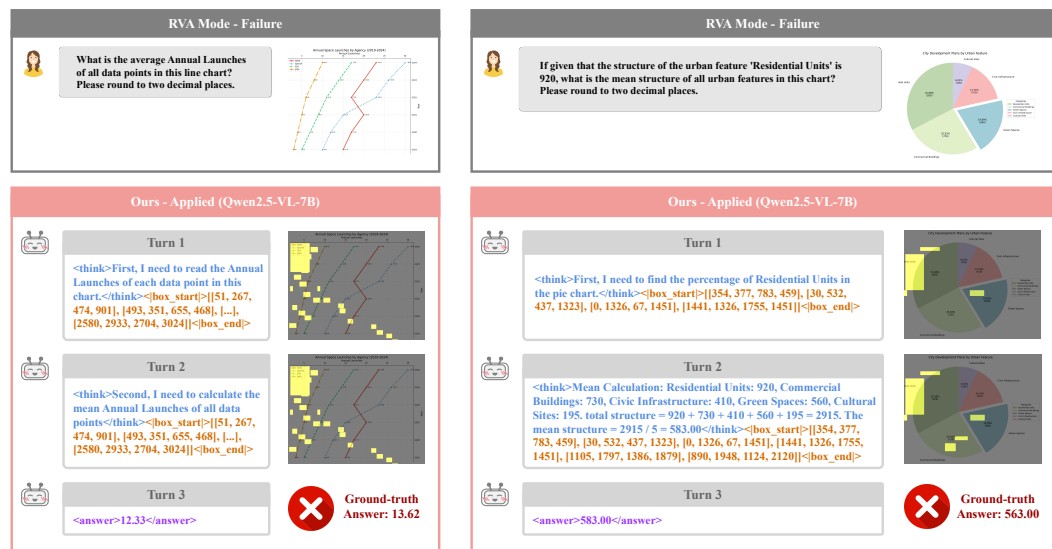

Figure 32: **Failure Examples Through Mode RVA.** This figure shows two examples of RVA inference on CCQA, where the model fails to give correct answers.

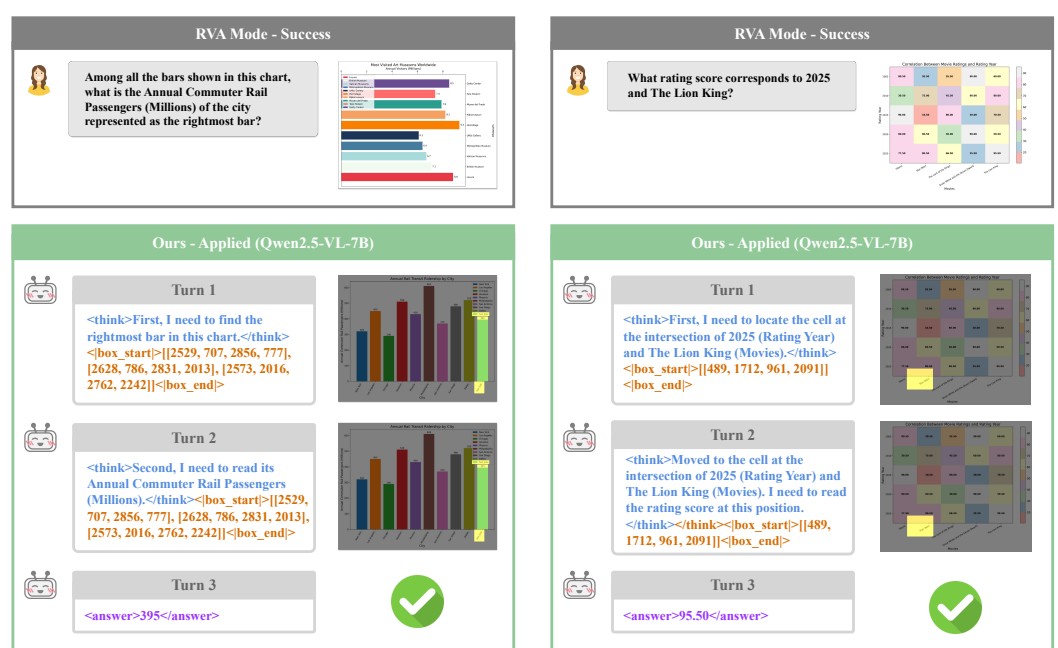

Figure 33: **Success Examples Through Mode RVA.** This figure shows three examples of RVA inference on CCQA, where the baseline Qwen2.5-VL-7B fails while our ✿CURV using *applied* grounding succeeds.

