# OpenReview forum: "CURV: Enhancing Chart Understanding through Visual Grounded Reasoning"
_ICLR.cc/2026/Conference — ICLR 2026 Conference Withdrawn Submission_

### Official Review · Reviewer_sseA · 2025-10-17

**Soundness:** 1
**Presentation:** 3
**Contribution:** 1
**Rating:** 4
**Confidence:** 5

**Summary:**

This paper addresses the poor performance of Multimodal Large Language Models (MLLMs) on Chart Question Answering (CQA). The authors posit that MLLMs lack "intrinsic" visually grounded reasoning capabilities. To address this, they propose two main contributions:
1) CURV, a curriculum learning framework that trains MLLMs to internalize reasoning;
2) CCQA, a new, scalable, synthetic dataset built with a three-level curriculum of increasing difficulty.

The CURV method uses Supervised Fine-Tuning (SFT) to train models on (Question, Reasoning, Bounding-Box, Answer) quadruplets. This process is designed to force the model to explicitly coordinate its logical reasoning steps ($R_t$) with dynamic visual grounding via bounding boxes ($B_t$).

**Strengths:**

1. Clear Problem Identification: The paper does a good job of identifying and demonstrating a clear failure mode in modern MLLMs: the disconnect between their reasoning chains and the visual evidence in an image.

2. Good Presentation: The paper is easy to follow.

**Weaknesses:**

1. Reliance on SFT is a Methodological Step Backward: The paper's entire framework is built on SFT. For a complex reasoning task, this is a major limitation. SFT is "imitation learning"; the model is only trained to mimic the single "golden path" reasoning trace provided by the dataset's templates. This approach is notoriously brittle, suffers from exposure bias (it never learns to recover from its own errors), and does not teach the model a generalizable policy for reasoning. The state-of-the-art in complex reasoning has moved decisively towards RL and preference-based methods, which are more robust. This paper's reliance on SFT feels outdated.
2. Trivial "Tools": The paper presents "applied," "boxed," and "cropped" as three distinct grounding strategies. "Applied" (a yellow highlight) and "Boxed" (a red border)  are functionally identical. "Cropped" (a zoom-in) is a standard, non-novel technique in computer vision. Presenting these three minor variations as a "comparative analysis" (Table 6)  further pads the paper without adding real substance. They both take bounding box coordinates and overlay a visual cue. The core mechanism is simply "training with bounding boxes."
3. Quality of Dataset: The dataset is presented as a core contribution, but its quality is highly questionable for training robust models. The dataset is synthetic and template-based. The semantic richness of the domain is likely ignored, as the task only requires applying a logical template to the data. Through SFT, the model might only memorize the template of the constructed dataset. It is a large, clean, but ultimately sterile and over-simplified dataset. Its template-based nature creates a "toy world" where reasoning is unrealistically clean and follows predefined paths. If a real-world problem requires a slightly different logical step, the model has never been exposed to it and will likely fail. This is not robust reasoning; it is high-fidelity mimicry.
4. Core Method is a Combination of Existing Ideas: The central framework, CURV, is not a new method. It is a straightforward combination of two very well-established concepts: (1) Curriculum Learning  and (2) Interleaved Reasoning. The idea of training models to produce step-by-step reasoning grounded in the image has been explored before. This paper simply applies a standard curriculum to this data format. This represents a solid engineering effort but lacks the fundamental research novelty expected at ICLR.

**Questions:**

1. Can you justify the decision to use SFT, an imitation-learning approach, over more modern RL-based methods for training a robust reasoning policy? How can you be sure your model is learning to "reason" rather than just "memorizing" the synthetic reasoning templates?
2. Can you defend the "meta-learning"  claim? How is your data generation process (systematically combining 30 domains, 7 chart types, and 12 operators)  fundamentally different from a standard, large-scale, and well-structured curriculum design?
3. To isolate the contribution of your method from your data, did you run a crucial baseline where a model is fine-tuned on the CCQA dataset (Levels 1+2) using only the final (Question, Answer) pairs, without the intermediate ($R_t, B_t$) supervision? This would quantify the true benefit of the grounded reasoning curriculum.

---

> ### Author Response · Authors · 2025-11-24
>
> Thank you very much for your valuable comments and constructive insights. We are very honored to know that you find our work to be well-motivated and easy to follow. We also very much appreciate your suggestions on extending our finetuning strategies beyond SFT, and we will include more experiments in our revision.
>
> Also thank you for your very insightful questions:
>
> (1) We initially choose SFT for its computation efficiency, as we find that finetuning 7B models through RL will require 80G×8 GPUs. While considering the robustness of RL-based methods, we are conducting experiments through RL finetuning, and will include more results into our revision.
>
> (2) Thank you for noting our “meta-learing” implementation. As we aim to help the model adapt to more complex CQA tasks and generalize to different domains beyond chart understanding, we think the core is to let the model learn to address complex problems as how humans would usually do, like decomposing complex problems to simpler steps, identifying visual substructures, analyzing visual information based on their elementary components, connecting logical reasoning with visual perception, etc. To support this, CCQA is built from a factored data-generation process that systematically combines a small set of chart types, domains, and reasoning operators. As such, it exposes the model to a controlled set of primitives and to many ways these primitives can be recombined. The result is not just a curriculum of increasing difficulty, but a training regime that encourages the model to infer general rules for assembling solutions to novel tasks. So this makes CCQA different from a standard large-scale curriculum dataset that covers a wide range of types, domains, formats, and complexities. Such expansion can notably increase variety but is unable to teach the model reusable patterns of reasoning and generalizable visually grounded reasoning capabilities. In contrast, our meta-learning formulation restricts the base elements while maximizing their systematic combinations, enabling the model to acquire abstractions that can transfer to unseen data and domains. On the other hand, a highly heterogeneous curriculum can inadvertently entangle domain-specific cues with reasoning patterns, making it harder for the model to identify generalizable strategies. We hugely thank for this insightful question, and will include more discussions in our revision.
>
> (3) Yes, we very much appreciate this valuable suggestion, and we will update the table shortly, along with extended results into our revision.
>
> Thank you again for your valuable comments and insightful suggestions.

---

### Official Review · Reviewer_oBSm · 2025-10-28

**Soundness:** 4
**Presentation:** 3
**Contribution:** 4
**Rating:** 8
**Confidence:** 5

**Summary:**

This paper addresses Chart Question Answering (CQA), where multimodal LLMs often fail to reason with visual grounding. The authors propose CURV, a curriculum learning framework that reformulates CQA as multi-turn reasoning with dynamic visual grounding, pairing each reasoning step with a bounding box in the chart. CURV is trained with a multi-objective loss combining language modeling and mask-based visual grounding.

To support training, they introduce CCQA, a synthetic dataset covering seven chart types and 30 domains, designed with progressive difficulty, interleaved visual grounding, and template-based accuracy. Experiments on CCQA, real-world, and out-of-domain benchmarks show CURV improves answer accuracy, reasoning, and visual grounding, with applied masking performing best.

**Strengths:**

- **Novel and well-motivated framework:** CURV addresses a clear limitation in MLLMs by internalizing visually grounded reasoning through multi-turn structured reasoning.
- **Effective curriculum learning:** CCQA is well-designed to support progressive learning, with strong alignment between reasoning and visual grounding.
- **Comprehensive evaluation:** Experiments cover multiple models, benchmarks, and metrics, including reasoning, visual grounding, and generalization.
- **Transparent analysis:** The authors honestly discuss trade-offs in curriculum levels, providing insight into robustness vs. generalization.

**Weaknesses:**

- **Limited CCQA dataset details:** The paper does not provide information on the total number of samples, nor the distribution of examples across the curriculum levels, which is important for assessing dataset scale and difficulty.
- **Relevant prior work missing:** The paper does not cite **[arXiv:2506.11991](https://arxiv.org/pdf/2506.11991)**, which also explores multimodal chain-of-thought reasoning and evaluation on charts; including this reference would strengthen the positioning.
- **Reproducibility:** There is no public code or repository for CURV or CCQA, which limits reproducibility of results.
- **Position of Related Work:** The Related Work section appears after Results; integrating it earlier could improve narrative flow and clarity.
- **Minor writing issue:** The sentence “These reveal MLLMs’ lack of logical decomposition and visual reasoning capabilities” has a typo.

**Questions:**

**Actionable Feedback**

1. Include CCQA statistics (number of samples, distribution across levels) to improve clarity.
2. Cite relevant prior work ([arXiv:2506.11991]) to strengthen contextual positioning.
3. Consider releasing code and CCQA dataset to support reproducibility.
4. Reorganize Related Work section to appear earlier for better narrative flow.
5. Fix minor writing issues, e.g., _“These reveal…”_ → _“These results highlight…”_.

---

> ### Author Response · Authors · 2025-11-24
>
> Thank you very much for your valuable comments and constructive insights. We are very honored to know that you find our work to be novel and well-motivated. We also greatly appreciate your insightful suggestions on improving our data statistics, extending related work, and avoiding writing typo. We will also make our code and data public in the near future.
>
> We also hugely thank your valuable suggestions on explaining our CCQA data statistics in further detail. Our dataset construction method is scalable to generate more data, while our current dataset is as follows:
>
> | Chart Type | Sour Image Num. | CQA Num. | Single-Chart CQA | Multi-Chart CQA
> |-----------|------------------|----------|-------------------|-------------------|
> | Bar       | 30               | 21254    | 16654 | 10200
> | Histogram | 30               | 13630    | 14622 | 10200
> | Scatter   | 30               | 28110    | 23110 | 10200
> | Line      | 30               | 22020    | 17020 | 10200
> | Heatmap   | 30               | 16340    | 17340 | 10200
> | Pie       | 30               | 17000    | 20000 | 10200
> | Radar     | 30               | 13428    | 14058 | 10200
> | **Total** | **210**          | **131782** | **122804** | **71400**
> | **Train** | **210**          | **20000**  | **20000**  | **0**
> | **Test**  | **210**          | **4200**   | **2800** | **1400**
>
> Thank you for your very insightful questions:
>
> (1) In addition to the table above, we will also add fine-grained dataset analysis to support our CCQA benchmark.
>
> (2) We will include both your suggested work and relevant prior work to strengthen different aspects of our work.
>
> (3) Thank you for your suggestion on this, and we will be sure to do so.
>
> (4) We will both enrich and better organize our related work section to improve our narrative flow.
>
> (5) We greatly appreciate your suggestions on this, and we will double check to eliminate writing issues.
>
> Thank you again for your valuable comments and insightful suggestions.

---

> > ### Comment · Reviewer_oBSm · 2025-11-26
> >
> > Thank you for addressing my comments. I encourage the authors to consider providing anonymized code and data in future submissions, which would help reviewers more easily reproduce and validate the proposed approach.

---

> > > ### Author Response · Authors · 2025-11-27
> > >
> > > Thank you so much for your thoughtful comments and insightful suggestions! We truly appreciate your valuable feedback. We will make the code and data for this work publicly available, and we will also setup anonymized links beforehand in future submissions to support easier reproducibility and validation.

---

### Official Review · Reviewer_eU2M · 2025-10-31

**Soundness:** 2
**Presentation:** 2
**Contribution:** 2
**Rating:** 2
**Confidence:** 4

**Summary:**

The paper introduces CURV, a curriculum learning framework designed to improve multimodal large language models’ ability to perform visually grounded reasoning for chart question answering (CQA). Instead of relying on external prompts or cues, CURV trains models to conduct multi-step reasoning while dynamically focusing on relevant chart regions. To support this, the authors construct CCQA, a synthetic curriculum dataset covering seven chart types and 30 domains, organized into three levels that range from simple single-operation tasks to multi-chart reasoning. Experiments on CCQA and multiple public benchmarks demonstrate that models fine-tuned with CURV achieve higher accuracy and better generalization than baseline models, suggesting that integrating structured visual grounding with stepwise reasoning enhances chart understanding performance.

**Strengths:**

The motivation is well defined. Sections 2.1 and 2.2 clearly explain why this task is important and highlight where current models are struggling.

The CCQA dataset hierarchy is well structured and effectively illustrated. The authors provide strong examples demonstrating where model reasoning fails for each hierarchy type. This hierarchical split is useful for evaluating how and where models struggle across different reasoning levels.

The three proposed strategies are clearly defined and well supported with examples. However, these explanations would be more impactful if they were discussed in greater detail in the main content rather than being confined to the appendix.

**Weaknesses:**

The CCQA dataset is not explained properly. While the paper mentions 7 chart types and 30 domain categories, implying 210 base images with “hundreds” of derived questions per image, the total number of questions is never specified. This makes it very difficult to evaluate this dataset. Additionally, the process for ensuring the correctness of the “query–reasoning–grounding–answer” quadruplets for each question is unclear and requires further explanation.

There is a lack of clarity regarding the baseline setup. The paper should elaborate on how prompting was done for the baseline, what visual or contextual information was provided alongside the images, and how this aligns with the proposed strategies. It’s not fully clear if baseline methods were given equally strong training or prompts, so comparisons may favor the new method.

Table 2 does not include boxed and cropped results for InternVL—these values are only shown for Qwen2.5VL. The reported performance improvements are minimal (1–4% overall increase in Table 3), which is expected given the fine-tuning setup. Even the 10.49% gain for Qwen2.5VL is limited to Level 1 and remains about 5% lower than GPT-4.1-mini. Without disclosure of the total question count, it is difficult to assess the statistical significance of these improvements. Furthermore, Table 3 omits closed-source models such as GPT-4o and GPT-4.1-mini.

The training starts from a small set of base charts that are reused a lot. This could introduce synthetic bias risk. Performance drops sharply when multiple subplots must be reasoned about together showing that the approach doesn’t work well for multi-chart settings. Few ablations on of how sensitive results are to number of steps, image resolution, etc.

The related work section is insufficient. It does not clearly articulate how this method differs from prior chart visual grounding approaches, such as RefChartQA, or from visual grounding methods applied to non-chart data. How is your method different from these existing grounding strategies? Additionally, it fails to properly distinguish CCQA from existing datasets mentioned in Table 3, which weakens the justification for the novelty of the proposed dataset.

Many results are judged by another AI, there’s limited human evaluation and no error bars or confidence intervals.

While the paper contains substantial and valuable content, the overall organization and ordering could be improved. Many important details are buried in the appendix, making it harder to follow the main narrative. Moving key explanations and examples into the main text would greatly improve readability and coherence.

**Questions:**

Section 2.1 mentions human-annotated visual information when discussing bottlenecks in MLLMs. How were these annotations performed, and were there any specific guidelines or quality-control procedures followed during the annotation process?

Since human-curated reasoning during inference is unfeasible, how different would the performance in Table 1 be if the VA and RVA setups used MLLM-generated visual information instead of human-annotated information?

Section 6.4 mentions that adding Level 3 training improves complex reasoning but weakens performance on simpler levels. How would changing hyperparameters such as w1,w2,w3​ (Formula 29), λ (Formula 25), and μ(i) (Formulas 27 and 28) affect this trade-off? What specific values of w(i) were used, and what was the rationale for those choices?

Given that the paper shows multi-chart understanding remains challenging, how does the proposed agent design or reasoning mechanism help address this issue?

How were the baselines setup? Can we check robustness of the approach when there are messy charts with lots of clutter and information overload?

How are regions chosen across multiple charts? How to prevent model from focusing on wrong regions or getting confused between multiple charts?

Can we show quality of grounding as a table to measure how often the model points to the right place and how that ties to answer accuracy?

Any small human review to check the reasoning and highlighted regions?

---

> ### Author Response · Authors · 2025-11-24
>
> Thank you very much for your valuable comments and constructive insights. We are very honored to know that you find our work to be well-motivated. We also very appreciate your insightful suggestions on moving important details into the main content, and we will reorganize them in our revised version.
>
> We also hugely thank your valuable suggestions on explaining our CCQA data statistics in further detail. Our dataset construction method is scalable to generate more data, while our current dataset is as follows:
>
> | Chart Type | Sour Image Num. | CQA Num. |
> |-----------|------------------|----------|
> | Bar       | 30               | 21254    |
> | Histogram | 30               | 13630    |
> | Scatter   | 30               | 28110    |
> | Line      | 30               | 22020    |
> | Heatmap   | 30               | 16340    |
> | Pie       | 30               | 17000    |
> | Radar     | 30               | 13428    |
> | **Total** | **210**          | **131782** |
> | **Train** | **210**          | **20000**  |
> | **Test**  | **210**          | **4200**   |
>
> During model training, we randomly sampled 10,000 CQA data from each level, and then randomly sampled 1,400 samples (nonoverlapped with training data) for evaluation.
>
> We also appreciate your valuable comments on baseline setup, and will extend our evaluation to more models. We will also extend our related work to include more valuable references. In addition to existing rubric evaluation and LLM-as-a-judge assessment, we will also include human evaluation to support our findings and observations. We apologize for putting some important details into the appendix, and we will reorganize in our revision to improve our presentation.
>
> Thank you for your very insightful questions:
>
> (1) Yes, we have a general annotation guideline during annotation to ensure annotation quality. We will include detailed explanations of this part in our revision.
>
> (2) In Table 1, we aim to identify what obstacles affect the model’s chart understanding performance. Therefore, we add CoT prompts (i.e., directly prompting the model to generate intermediate reasoning steps before giving the final answer) and human-annotated textual visual descriptions to see if they may take effect. In Table 2, we aim to validate our visual grounding method, so we directly use our benchmark data, which contains concrete reasoning steps (different from prompting the model to do reasoning in Table 1) and visual grounding images. As MLLM-generated visual information is a major cause of CQA failure, we also observe impaired performance when directly using MLLM-generated visual descriptions. We will add concrete results for this in our revision.
>
> (3) Thank you for noting this detail, and we used the same hyperparameter configuration during training and evaluation. We set $w_1=0.6, w_2=w_3=0.2$, $\lambda=0.5$, and $$\mu_2=\mu_3=0.8$. We will extend our Table 7 to explain hyperparameter settings in further detail.
>
> (4) We aim to train the model to be adaptive to more complex CQA tasks, and generalizable to different domains by simulating human reasoning patterns. So we decompose the complex problems into simpler sub-steps. In this way, the model is able to tackle more complex problems by going through more intermediate simpler reasoning steps.
>
> (5) We thank the valuable suggestions on baseline setup and robustness assessment, and will add more details on these in our revision.
>
> (6) Regions are randomly selected during single-chart and multi-chart localization tasks. For multi-chart relation understanding tasks, we predefine a set of relations, which are then randomly selected to specify the queried regions. Your question is also what we aim to tackle, and we will include more experiments on this valuable point.
>
> (7) Yes, we will definitely include this analysis as a table, which will be of huge help to understand model failure patterns and potential future improvement directions.
>
> (8) Yes, we have randomly selected samples to validate our dataset quality. We will include quantitative results to support our assessment.
>
> Thank you again for your valuable comments and insightful suggestions.

---

> > ### Comment · Reviewer_eU2M · 2025-11-26
> > **Thanks for you Response**
> >
> > Thank you for your responses and clarifications. After considering your rebuttal, I have decided to improve my original score for the submission.

---

> > > ### Author Response · Authors · 2025-11-27
> > >
> > > Thank you so much for your thoughtful consideration and insightful comments! We very much appreciate your decision to raise our score, and we will carefully revise our paper to improve its organization and include more results.

---

### Official Review · Reviewer_URig · 2025-11-02

**Soundness:** 3
**Presentation:** 3
**Contribution:** 3
**Rating:** 2
**Confidence:** 4

**Summary:**

The paper proposes CURV, a curriculum learning framework designed to build intrinsic visual grounded reasoning capabilities in models. This is achieved by progressively training models to coordinate visual attention with logical reasoning, starting with simple single-operation tasks and moving toward complex nested reasoning.

**Strengths:**

- Novelty and Technical Soundness: The idea of using grounded visual reasoning is novel and technically sound, providing a strong approach to improve performance at the intersection of LVLMs and structured data understanding.
- Clarity: The paper is well-written and easy to follow, with a clear structure.
- Motivation: The motivation behind the work is interesting and well-presented.
- Novel Finding: The finding in the analysis that MLLMs can internalize these capabilities through grounded visual reasoning is novel.

**Weaknesses:**

## Dataset Quality and Assessment (Major Concern)

The quality and statistics of the proposed dataset are not well-assessed. The majority of the dataset's details are discussed in the supplementary material, and even after reading through it, the reviewer cannot find an in-depth analysis of the proposed dataset. Since this paper heavily relies on the proposed dataset, a thorough analysis and detailed statistics are mandatory to warrant the effectiveness of the methods.
- Required Data Statistics: The authors should provide essential data statistics, such as the number of training and testing data samples, and other distributions like question types.
- Bias and Diversity: Based on the supplementary material, the dataset is generated by an LLM, which can introduce bias (e.g., a monotonic trend thatmostly upward or downward in generated samples) and may lack the diversity required to cover real-world distributions. Does the data pipeline face similar issues? How is this issue mitigated or solved?
- Quality Control: Since the LLM can make mistakes, is there any post-filtering or human-in-the-loop filtering involved in the dataset generation?
- Quality Assurance: To ensure the quality of the dataset, can the authors perform a human evaluation on both the training and test sets to demonstrate the correctness and diversity of the synthetic data?
- Context: How many data samples are in the training set and testing set? Please include a comparison with previous datasets or benchmarks.

## Effectiveness Assessment and Experimental Comparison (Major Concern)

The effectiveness of the dataset and method is not well-assessed in the experimental comparisons (Tables 2 and 3). The current comparison involves models fine-tuned on chart-specific data against their baseline models on a chart benchmark. It is obvious and not surprising that fine-tuning on a specific downstream domain will improve the domain capability of general models like QwenVL and InternVL. A more head-to-head comparison should be considered: For example,
- To Assess Dataset Effectiveness: It is suggested to have a baseline model trained exhaustively on existing chart domain training data (e.g., the 1.4M training data organized by TinyChart) and compare it with the same baseline model trained on both classical chart domain data plus the proposed data. Assuming the classical data has reached its limit to boost performance, if the proposed dataset can bring significant further improvement, one would be convinced of its effectiveness.
- To Assess Visual Grounding: To assess the benefit of visual grounding reasoning training, compare a model trained with and without the grounding loss on the exact same training dataset. This way, with the training data fixed, the actual benefit from the grounding loss training can be clearly observed.

*The reviewer acknowledges this is a long paper with many details in the supplementary material and may have missed specific points mentioned in the appendix. They are willing to reevaluate the paper if this concern can be addressed.*

**Questions:**

- In line 81, the authors mention "coordinate visual attention" and a "mask-based grounding loss." How exactly are the masks collected? And how is the mask loss computed.is it based on the bounding boxes proposed in the reasoning process or the output-to-vision tokens attention map?
- How is the human-annotated visual information provided to the model? Is it provided exactly as a set of bounding box coordinates?
- The trends observed in Table 1 and Table 2 appear to be opposite. Table 1 suggests that human-annotated visual information brings significantly more value than CoT reasoning for chart domain understanding. However, this observation does not hold in Table 2, where the opposite trend is seen: reasoning (blue bar) brings more improvement than visual information (yellow bar). Can the authors provide more discussion on this discrepancy?

---

> ### Author Response · Authors · 2025-11-24
>
> Thank you very much for your valuable comments and constructive insights. We are very honored to know that you find our paper to be novel and well-presented. We also highly agree with your suggestions on in-depth dataset analysis and ablation studies. We are conducting additional experiments and will add new results to our revised version. Besides our Figs. 6 and 10, we will also add additional data statistics and quality assessment.
>
> Our training and testing data are as follows, randomly sampled from CCQA, with training and test sets nonoverlapping with each other:
> | Curriculum Level | Train  | Test |
> |------------------|--------|------|
> | 1                | 10000  | 1400 |
> | 2                | 10000  | 1400 |
> | 3                | -      | 1400 |
>
> Thank you for your very insightful questions:
>
> (1) We collected our masks during the chart figure generation phase, during which we first draw the original chart figure, and then mask the specific elements using matplotlib functions. By comparing the original chart and the masked chart, we can obtain the binary mask and the corresponding bounding box coordinates. As we observe improved performance through reinforcement learning, we will add concrete experiments to explain the loss and reward for supervised finetuning and reinforcement learning.
>
> (2) Human-annotated visual information is given during the figure generation phase, where human annotates the visual elements as the queried visual grounding, followed by the auto-generated binary masks and bounding boxes.
>
> (3) In Table 1, we aim to identify what obstacles affect the model’s chart understanding performance. Therefore, we add CoT prompts (i.e., directly prompting the model to generate intermediate reasoning steps before giving the final answer) and human-annotated textual visual descriptions to see if they may take effect. In Table 2, we aim to validate our visual grounding method, so we directly use our benchmark data, which contains concrete reasoning steps (different from prompting the model to do reasoning in Table 1) and visual grounding images. So the differences between prompting CoT and concrete CoT, together with textual visual description and visual grounded images, lead to the differences in performance improvements. We apologize for this confusion, and will add clearer explanations to our preliminary exploration to better support our benchmark and method.
>
> Thank you again for your valuable comments and insightful suggestions.

---

### Author Response · Authors · 2025-12-02

We sincerely thank all the reviewers for their valuable comments and insightful suggestions. Your feedback has been highly constructive, and we have already begun enriching our work with additional training and evaluation results. However, due to computational limitations, we regret that we may not be able to complete all the additional experiments by the deadline. With this in mind, we have made the difficult decision to withdraw our submission.

We are truly grateful for all the constructive feedback we received, and we will carefully incorporate it as we continue to improve our work for future revisions. Sincerely appreciate your time and efforts!

---

### Note · Authors · 2025-12-02

**Comment:**

We sincerely thank all the reviewers for their valuable comments and insightful suggestions. Your feedback has been highly constructive, and we have already begun enriching our work with additional training and evaluation results. However, due to computational limitations, we regret that we may not be able to complete all the additional experiments by the deadline. With this in mind, we have made the difficult decision to withdraw our submission.

We are truly grateful for all the constructive feedback we received, and we will carefully incorporate it as we continue to improve our work for future revisions. Sincerely appreciate your time and efforts!

**Withdrawal Confirmation:**

I have read and agree with the venue's withdrawal policy on behalf of myself and my co-authors.